# CELF: A Self-Supervised Multimodal Framework for Concept-Based Interpretability

## Abstract

Traditional XAI techniques in computer vision, such as heatmaps and saliency maps, highlight input regions that influence model predictions. However, they often lack precision and may introduce bias. Concept-based XAI approaches, such as concept bottleneck models or textual explanations of latent neurons, aim to provide more interpretable representations but typically rely on human-annotated concept sets, which are scarce in specialized domains. Moreover, Large Language Models (LLMs) used for automatic concept generation can hallucinate, reducing reliability and trust. To address these challenges, we propose the Concept Extraction and Learning Framework (CELF), a self-supervised multimodal method for extracting and retrieving human-interpretable concepts from vision-language data without manual annotations. CELF integrates attention-guided keyphrase extraction with contrastive learning, and applies a graph-guided refinement stage to promote semantic consistency. For controlled evaluation, we introduce C-MNIST, a configurable dataset generator with ground truth concepts. Experiments on C-MNIST, Visual Genome, and CUB demonstrate that CELF outperforms prior baselines in concept extraction and improves multi-label classification performance on C-MNIST.

## 1 Introduction

Deep learning models are increasingly deployed in high-stakes domains such as healthcare, autonomous systems, and scientific discovery, where decisions must be reliable and interpretable. Yet, these models often operate as black boxes, lacking transparency into their decision-making processes (Akhai, 2023; Das & Rad, 2020). While eXplainable Artificial Intelligence (XAI) seeks to mitigate this issue, many traditional methods suffer from trade-offs between interpretability and performance, imprecise saliency, and a risk of introducing bias-induced misinterpretation (Akhai, 2023; Yuksekgonul et al., 2023).

In contrast, concept-based XAI offers a promising alternative by associating decisions with human-interpretable concepts, such as "red," "striped," or "wings." These methods include Concept Bottleneck Models (CBMs) and techniques that align neurons or features with textual concepts (Lee et al., 2024; Poeta et al., 2023). While effective, these approaches typically require manually labeled concept datasets, which are scarce and expensive to produce (Alzubaidi et al., 2023; Lee et al., 2024; Poeta et al., 2023). Recent work has proposed using Large Language Models (LLMs) to generate concepts automatically (Gao et al., 2024; Yan et al., 2023b). However, LLMs frequently produce incomplete, biased, or hallucinated outputs (Patil et al., 2024; Wang et al., 2025), raising concerns about their reliability and consistency for concept extraction. In addition, recent approaches leverage Vision Language Models (VLMs) and cross-modal alignment to extract concepts. However, they often rely on large-scale pretraining and assume prior knowledge of the dataset's concept space or domain structure.

In response, this work makes three main contributions:

1. Concept Extraction and Learning Framework (CELF): a self-supervised multimodal framework that extracts and aligns interpretable visual concepts from image–text pairs without manual concept labels. It uses concept-guided attention and contrastive supervision to group semantically coherent textual fragments and align them with relevant image regions. CELF improves robustness, scalability, and interpretability, and outperforms baselines in both

concept extraction and a downstream task on synthetic and real-world datasets with ground truth concepts.

2. Conceptual-MNIST (C-MNIST): A flexible data generator providing ground truth concepts in visual and textual forms. C-MNIST enables scalable, reproducible evaluation of concept-based XAI methods.

3. Semantic Cosine Similarity (SCS): A conservative evaluation metric for concept extraction that captures semantic matches beyond exact phrase overlap, addressing the limitations of rigid lexical metrics and the overly permissive nature of contextual ones like BERTScore.

## 2 RELATED WORK

**Concept Extraction (CE) & Concept Retrieval (CR).** A concept is an abstraction represented by human-interpretable features, expressed through modalities like text or visual regions (Lee et al., 2024; Poeta et al., 2023; Schwalbe, 2022). In this work, we focus on two core tasks of concept-based XAI: CE and CR. CE identifies relevant concepts within a dataset without predefined annotations, whereas CR matches these concepts with the corresponding image features in new image-only samples.

Table 1 compares representative methods across CE and CR tasks, highlighting their strengths and limitations. We focus our discussion on VLG-CBM (Srivastava et al., 2024), XCB (Alukaev et al., 2023), FALCON (Kalibhat et al., 2023), and Grad-ECLIP (Zhao et al., 2024). We select these methods because FALCON, XCB, and Grad-ECLIP are closely aligned with our experimental and evaluation settings, while VLG-CBM serves as a representative example of a CBM that relies on LLMs to generate concepts. Additional related works are discussed in Appendix A.

- **FALCON** (Kalibhat et al., 2023) identifies highly activating regions in an encoder, uses Contrastive Language-Image Pre-training (CLIP) to retrieve captions for the cropped regions, and extracts candidate concepts (e.g., nouns or noun phrases) based on similarity scores. It then applies contrastive filtering to retain only the most relevant concepts. While it requires no training and provides a simple pipeline, it heavily depends on the quality of CLIP's captions and noun phrase extraction, which can result in shallow concept sets and poor generalization across domains.

- **XCB** (Alukaev et al., 2023) learns concept queries directly from image-text pairs without predefined concept labels. However, it uses a fixed concept set and requires supervision from a downstream classification task, limiting its flexibility, especially in zero-label settings.

- **VLG-CBM** (Srivastava et al., 2024) leverages LLMs to generate concepts per class, followed by filtering strategies, such as grounded filtering, to remove irrelevant or noisy concepts, producing image-level annotations.

- **Grad-ECLIP** (Zhao et al., 2024) computes gradients of the CLIP similarity score with respect to both image patches and text tokens, generating saliency maps for both modalities. This provides fine-grained visual and textual attributions but lacks structured concept representation. It is also sensitive to gradient noise and tokenization artifacts (e.g., subwords), which hinder reliable word-level interpretation and, by extension, concept-level understanding.

**CLIP Modifications for Fine-Grained Alignment.** Modifications to CLIP (Radford et al., 2021) have explored enhancing fine-grained visual-text alignment (Huang et al., 2021; Wang et al., 2022), adjusting training objectives, adding layers (Fu et al., 2022; Mu et al., 2022; Yao et al., 2022; Zhao et al., 2023), and integrating textual and visual concepts (Liu et al., 2021; Zhang et al., 2024). Yet, these methods often lack mechanisms for structured CE. CELF addresses this gap by integrating LLM-generated pseudo-labels with CLIP's multimodal learning, enabling scalable CE and CR.

**Synthetic datasets for concept-based XAI.** Synthetic datasets can overcome the lack of annotated concepts. However, existing ones focus on visual concepts with fixed configurations, limiting flexibility for multimodal evaluation, particularly in self-supervised scenarios (Posada-Moreno et al., 2024; Yeh et al., 2020). While some works explore textual concepts, they often suffer from a limited number of concepts and unnatural-sounding captions (Alukaev et al., 2023). Another line of work (Bader et al., 2025) generates synthetic images by applying diffusion-based attribute

Table 1: Comparison of CELF with other CE and CR methods, highlighting key differences and limitations. $\sim$ indicates partial CE, as Grad-ECLIP focuses on token-level interpretability.

| Method | CE | CR | Strengths | Limitations |
|---|---|---|---|---|
| **CELF** | ✓ | ✓ | Mitigates LLM hallucinations, learns structured concepts | May group semantically similar concepts |
| FALCON | ✓ | × | No training required, efficient subspace probing | Relies on CLIP similarity; requires rich captions |
| XCB | ✓ | × | No pretrained models needed, task-adaptive | Fixed concept set; requires downstream supervision |
| VLG-CBM | × | ✓ | Annotates images with a grounded object detector | Relies on the performance of a detector and LLM |
| Grad-ECLIP | $\sim$ | × | Provides visual and textual explanations for CLIP | Focuses on token-level scores |

substitutions to CUB images and filtering the outputs for consistency. Although this enables flexible dataset augmentation, the approach inherits diffusion-model biases, meaning concept correctness cannot be guaranteed in a rule-based manner.

To allow controlled multimodal evaluations with reliable annotations, we designed C-MNIST, a configurable generator that produces adaptable images, descriptions, and concept annotations with correctness guaranteed by construction.

## 3 CONCEPTUAL-MNIST (C-MNIST)

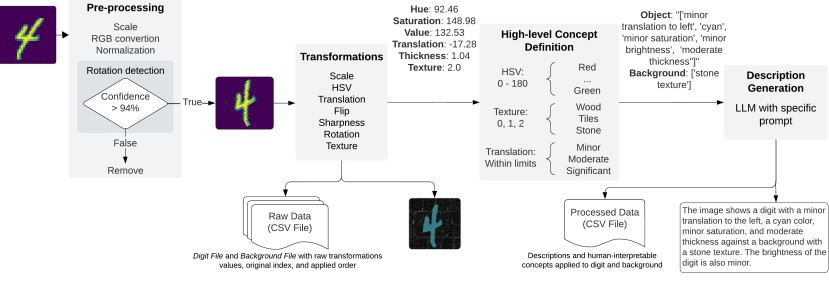

Figure 1: Example outputs at each stage of the C-MNIST generator. The process begins with image pre-processing, followed by transformations that yield modified images and CSV files. Continuous transformation values are converted into discrete textual concepts, which are used to prompt an LLM for generating descriptive image characterizations. See Appendix B for rotation detection and prompt.

To address the lack of annotated concept datasets for evaluating concept-based approaches, we introduce C-MNIST, a configurable data generator built on the MNIST dataset (Deng, 2012) and geometric shapes to produce datasets with controlled image-description pairs and corresponding concepts. While designed primarily as a benchmark for assessing concept extraction quality (and additionally for tasks such as retrieval, concept disentanglement, and compositionality), it is not intended to replace CBM-based evaluation on downstream accuracy.

Unlike previous static datasets (Posada-Moreno et al., 2024; Rosasco et al., 2024; Yeh et al., 2020), C-MNIST allows for dynamic configuration of concepts, offering greater flexibility and control in generating images and corresponding descriptions, with concept annotations.

Figure 1 illustrates the C-MNIST generation process:

- **Preprocessing:** Each MNIST image is resized to 256×256 pixels, converted to RGB, and normalized. This ensures consistency and supports color-based transformations. The digit's orientation is aligned as detailed in Appendix B.
- **Transformations:** Images undergo a set of customizable transformations, including continuous attributes (e.g., color, saturation, scale, translation, rotation) and discrete attributes (e.g., flip, texture). Users can specify the sequence of parameterized transformations or allow the system to apply them randomly.
- **High-level Concept Definition:** Transformed images are annotated with structured concepts. Continuous attributes are discretized for interpretability (configurable bins, e.g., "small scale").
- **Description Generation:** Textual descriptions are generated using the Zephyr 7B $\beta$ LLM [1] (Tunstall et al., 2024), guided by a prompt (Appendix D) that incorporates an example to ensure consistency.

The output comprises the transformed images, their descriptions, and a tabular dataset where columns represent the transformations, their values (e.g., 30º rotation), their sequence, and the original data indices to ensure reproducibility.

## 4 DATASETS

We evaluated CELF across both synthetic and real-world datasets. The former served as a controlled environment for validating CELF's performance in concept extraction and retrieval, and for supporting the selection of the specific approach to apply to real-world data. For consistency with HuggingFace's CLIP implementation, all images were converted to RGB, resized to 224×224 pixels, and normalized using the dataset's mean and unit standard deviation for all channels.

**Complex C-MNIST.** This dataset consists of approximately 65,000 training instances, 15,000 for validation, and 10,000 for testing. This size provides a balance between computational feasibility and training time, although larger datasets could further improve model robustness and generalization. The dataset includes images with 4–6 concepts per image, applied to both objects and backgrounds. Descriptions are occasionally incomplete and use synonyms instead of exact matches, reflecting real-world variation. With 37 concepts in total (see Appendix C.1), it tests CELF's ability to handle conceptual complexity and ambiguity.

**Visual Genome.** A widely used general-domain dataset with bounding box and attribute annotations, Visual Genome (Krishna et al., 2016) enables evaluation of concept extraction from real-world images with diverse, natural language descriptions (see more details in Appendix C.2). We treat object attributes as ground truth concepts. To generate caption–concept pairs, we group dissimilar region-level descriptions associated with annotated regions. These descriptions are embedded using a SentenceTransformer model (Reimers & Gurevych, 2019), and a subset of dissimilar phrases is selected and merged into a single caption, along with their corresponding concepts. This process is repeated to create multiple caption–concept pairs per image.

**CUB-200.** The CUB-200-2011 dataset (Wah et al., 2023) is a fine-grained visual classification benchmark containing images of 200 bird species (see more details in Appendix C.3). Each image is annotated with attributes describing physical features such as beak shape, wing color, and body pattern, making it ideal for concept extraction evaluation. For concept extraction from image–caption pairs, we adapted the approach in (Alukaev et al., 2023), treating attributes and their associated uncertainty terms as concepts. To ensure a fair comparison with image-focused methods, we excluded uncertainty terms like "not visible" and "maybe," retaining only "definitely" and "probably."

## 5 CONCEPT EXTRACTION AND LEARNING FRAMEWORK (CELF)

CELF is a self-supervised, multimodal framework designed to extract human-interpretable concepts from vision-language datasets without manual annotations and retrieve such concepts from new image samples. The framework, illustrated in Figure 2, operates in two core stages: CE and CR, leveraging LLMs and CLIP to refine concepts and accurately align them with visual features.

---

[1] See Appendix E for details on model selection, rationale, and implementation.

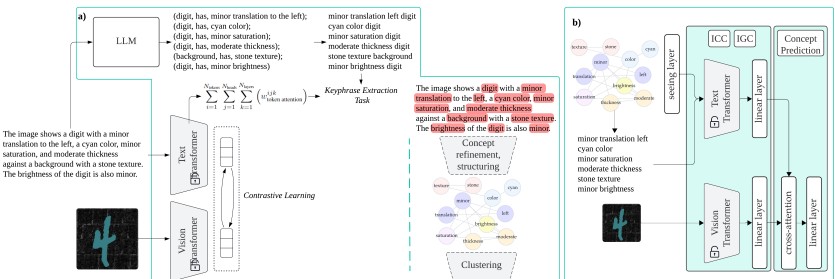

Figure 2: Example inputs and outputs at each stage of CELF: (a) CE – Pseudo-label generation for initial concept annotation, followed by CLIP-based extraction. A fine-tuned CLIP identifies relevant words, which are then refined and grouped into structured concepts (e.g., represented as colored circles). (b) CR – A modified CLIP is used to retrieve concepts and its application in downstream tasks. ICC and IGC stands for image-concept and image-graph contrastive learning, respectively.

## 5.1 CONCEPT EXTRACTION

The CE stage integrates the strengths of LLMs and CLIP to refine and extract reliable concepts. Individually, they have shown potential for CE (Gao et al., 2024; Kalibhat et al., 2023; Oikarinen et al., 2023; Yan et al., 2023b; Yang et al., 2023) but their combination remains less explored for caption-level concept extraction from image-text pairs. Our approach addresses the hallucination tendencies of LLMs and the noise susceptibility of CLIP.

### 5.1.1 PSEUDO-LABEL GENERATION

We generate pseudo-labels using the Phi-3 Mini LLM (Abdin et al., 2024), which has a 4,000-token context length, enabling rapid and contextually informed outputs. The model is prompted to extract structured triplets in the format (*object*, has, *concept*) from textual descriptions, inspired by the approach of Shi et al. (2023). Prompt engineering aimed to maximize domain alignment (see Appendix D).

### 5.1.2 CLIP'S FINE-TUNING

We use CLIP with a Vision Transformer (ViT) backbone, employing image-text contrastive learning to align image-description pairs by optimizing the symmetric cross-entropy loss over similarity scores (Radford et al., 2021). Importantly, image-text alignment allows CLIP to scale, even if the LLM fails to generate concepts correctly or encounters an unseen textual description during inference.

In addition to CLIP's contrastive loss, we introduce a novel keyphrase extraction task, framed as a multi-label classification problem. We train CLIP's text encoder to predict which words in the caption represent concepts, using a Binary Cross-Entropy (BCE) loss with pseudo-labels as supervision. To guide concept identification, we leverage CLIP's internal attention weights, where higher values denote stronger token relevance. The total attention weight of a token is defined as the sum across all layers, attention heads, and tokens, i.e. $w_{\text{token attention}} = \sum_{i=1}^{N_{\text{tokens}}} \sum_{j=1}^{N_{\text{heads}}} \sum_{k=1}^{N_{\text{layers}}} w_{\text{token attention}}^{ijk}$. For multi-token words, we average scores across their subword tokens to avoid bias toward longer words. During training, we apply min-max normalization to rescale scores into the [0, 1] range. This allows us to identify high-attention tokens without introducing additional learnable parameters. We chose not to add a learned linear head (e.g., an MLP or linear classifier) with a sigmoid output, as doing so would increase model capacity and risk memorizing noisy pseudo-labels. Instead, we rely directly on CLIP's representations and regularize fine-tuning by jointly optimizing CLIP's contrastive loss. This helps preserve the model's general image-text embedding structure while improving alignment with task-specific concepts and preserving transparent attention-based concept scores.

The total loss function combines the contrastive loss and BCE loss as follows: $L = \alpha L_{\text{CLIP}} + \beta L_{Keyphrase\ Extraction\ BCE}$, where $L_{\text{CLIP}}$ is the standard contrastive loss for aligning images and text, $L_{Keyphrase\ Extraction\ BCE}$ is the BCE loss for predicting concept-relevant words, and $\alpha$ and $\beta$ are hyper-

parameters controlling the trade-off between losses. Additional details about experiment setup can be found in Appendix F.1.

### 5.1.3 CONCEPT REFINEMENT AND STRUCTURING

Following the keyphrase extraction approach, we obtain scored words after normalization and a sigmoid function. At inference time, we use z-score normalization followed by a sigmoid to improve score consistency across inputs and datasets without requiring threshold tuning. We then select words surpassing an empirically determined threshold of 0.4.

A graph-based representation is constructed where nodes represent extracted words, and edges are determined in two steps: first, consecutive words in the pseudo-labels get higher edge weights; then, remaining weights are estimated using word pairs co-occurrence frequencies and mutual attention scores from CLIP's first four and last three layers. The attention scores from these layers provide insight into the contextual relevance between words, strengthening the connections between conceptually related terms.

Finally, the grouping of words into concepts begins with those identified in the pseudo-labels. Remaining words are grouped using Louvain community detection (Blondel et al., 2008), which identifies coherent concept structures based on graph connectivity.

**Bias and noise mitigation.** Using both pretrained CLIP and LLM-derived pseudo-labels introduces potential noise, which we mitigate through two complementary mechanisms: (1) only concepts that appear in the caption and receive sufficient CLIP attention are retained, preventing ungrounded LLM hallucinations from entering the training signal; and (2) semantically weak words, such as region-related terms (e.g., "digit," "shape," "background," "bird") and most stop words (Bird & Loper, 2004) (except negations due to their contextual significance), are removed before graph construction, which stabilizes Louvain clustering and prevents these tokens from dominating multiple concepts.

### 5.1.4 CONCEPT CLUSTERING

Synonyms can hinder CR, leading to false negatives, while a large number of distinct concepts significantly increases computational cost. To address this, semantically similar concepts are clustered to reduce redundancy and improve both performance and resource efficiency. Low-frequency concepts (occurrence $< 0.01\%$) are discarded as noise. Remaining concepts are embedded using a Sentence-Transformer model and reduced in dimensionality with Uniform Manifold Approximation and Projection (UMAP) (Allaoui et al., 2020). Clustering is performed using HDBSCAN (McInnes et al., 2017), with a secondary stage for low-confidence clusters to capture finer distinctions.

Finally, semantically equivalent clusters are merged when their centroids are highly similar, while less similar clusters are associated with broader parent clusters to preserve contextual relevance. Full implementation details, including embedding model, dimensionality, and hyperparameters, are provided in Appendix F.2.

## 5.2 CONCEPT RETRIEVAL

To address the localized nature of concepts in real-world vision-language datasets, we integrate a cross-attention module into CLIP. This module is inserted after the projection layers. In this setup, concepts are used as queries, and the image features act as both keys and values. The resulting attention outputs are token-averaged, normalized via a LayerNorm, and passed through a final linear layer. This design encourages each concept to attend to relevant spatial regions in the image.

### 5.2.1 FIRST STAGE: GRAPH-GUIDED REFINEMENT

We fine-tune CLIP using a graph-aware attention mechanism. Inspired by (Liu et al., 2020), we introduce a "seeing" layer that masks attention based on a concept graph, allowing only semantically connected concepts to interact. This promotes coherence and suppresses noise from unrelated concepts. The training optimizes a multi-objective loss composed of:

- **Image-Graph Contrastive Loss**: aligns image embeddings with their corresponding concept graphs, promoting structured concept understanding.

- **Image-Concept Contrastive Loss**: enforces fine-grained alignment between image regions and individual concept embeddings.

The final loss is a weighted combination of the two components: $L = \alpha L_{image-graph} + \beta L_{image-concepts}$, where $\alpha$ and $\beta$ are hyperparameters that control the relative importance of each loss. Details about experiment setup can be found in Appendix F.3.

### 5.2.2 SECOND STAGE: SUPERVISED CONCEPT PREDICTION

After the first stage, CLIP's weights are frozen. We then train the newly added layers using BCE loss to predict the presence of concepts in each image. The model outputs one score per concept, based on the token-averaged attended features, allowing for robust multi-label concept retrieval.

### 5.3 SEMANTIC COSINE SIMILARITY

In CE tasks, traditional metrics (e.g., precision, recall) are limited due to their rigid phrase-matching need, which does not account for synonyms or variations of ground truth concepts. As highlighted in (Papagiannopoulou & Tsoumakas, 2020), these metrics can yield overly pessimistic results, flagging predictions as incorrect even when they represent a subset, superset, or synonym of the ground truth concept. BERTScore (Zhang* et al., 2020), commonly used for text generation evaluation, can be used as an alternative, but its reliance on contextual embeddings may assign high similarity scores to concepts with similar contexts but different meanings.

To address these limitations, we propose a more conservative evaluation metric, SCS, which relies on embedding both the extracted and ground truth concepts. We adopt MiniLM-L6-v2 from Sentence-Transformers (Reimers & Gurevych, 2019) for the embedding task. This model was chosen for its strong performance on semantic similarity tasks, providing a robust distinction between true synonyms and semantically unrelated terms. Cosine similarity is computed between the embeddings, and a threshold of 80% is used to determine semantic matches. The threshold was empirically optimized to balance sensitivity and specificity, ensuring accurate evaluations without overestimating semantic proximity. The validation of SCS on controlled perturbations is provided in Appendix G.1.

### 5.4 CONCEPT-BASED CLASSIFICATION SETUP

To evaluate the contribution of extracted concepts to downstream tasks, we designed a classification setup using the complex C-MNIST dataset, which includes ground truth concept annotations. The goal was to determine whether extracted concepts provide useful semantic information for classification beyond what is captured by raw image features.

We used a pretrained ResNet50 to extract image features and concatenated them with concept features before the classification layer. The model predicts multi-label outputs corresponding to concept combinations, e.g. different position $\rightarrow$ concepts related to rotation and flip (see Appendix C.1 for label mapping details).

To evaluate the role of concept quality and retrieval, we compared two concept representations:

1. **Retrieved** concept embeddings: concepts selected based on the cross-attention CLIP retrieval mechanism.
2. **Extracted** concept embeddings: extracted concepts, bypassing the retrieval step. This provides an upper-bound for classification performance based on perfect concept retrieval.

## 6 RESULTS

In this section, we evaluate the performance of our method across three experimental settings: concept extraction, concept retrieval, and a downstream classification task. For concept extraction, the SCS metrics (precision, recall, and F1) are computed concept-wise, with F1 being the harmonic mean of the final recall and precision scores. BERTScore is applied by selecting the best match for each reference concept among the predictions and averaging these scores across all references in a sample. For both retrieval and downstream classification, we use macro-averaged metrics to evaluate concept/label prediction, providing a robust measure of performance under class imbalance.

## 6.1 CONCEPT EXTRACTION

Since no method is specifically tailored for concept extraction from image-caption pairs, we select baselines that operate directly on such pairs, enabling quantitative evaluation against caption-derived ground truth. We adapt FALCON* (Kalibhat et al., 2023) and GRAD-ECLIP (Zhao et al., 2024) for comparison. FALCON* originally uses a ResNet encoder and contrastive interpretability, which we modify by replacing the encoder with CLIP and disabling contrastive interpretability to suit our setting (details in Appendix G.3). GRAD-ECLIP is adjusted by applying spaCy for noun phrase extraction and empirically setting a 35th percentile threshold for filtering relevant terms.

In addition to these off-the-shelf baselines, we include three components from our pipeline: (1) CLIP-attn, our method without CE fine-tuning; (2) CELF, our method with CLIP fine-tuning, demonstrating its potential and enabling domain adaptation; and (3) LLM, which is used solely for pseudo-labeling and guiding CELF's concept refinement and serves mainly for ablation studies. For a fair comparison with off-the-shelf models, our primary focus is on CLIP-attn.

As shown in Table 2, CLIP-attn consistently performs well across datasets in terms of SCS-Recall. In C-MNIST, it achieves strong performance across all metrics relative to other baselines. While the LLM generates fewer concepts, they yield high precision; CLIP-attn achieves better recall, with lower precision due to occasional noise extraction. FALCON* underperforms because it relies on simple noun phrase extraction and is sensitive to caption variability, while GRAD-ECLIP struggles, likely due to domain shift and lack of filtering for irrelevant terms.

In Visual Genome, CLIP-attn leads in SCS metrics, while GRAD-ECLIP performs better than before due to alignment with CLIP's pretraining. FALCON* performs poorly, likely because retrieved captions tend to focus narrowly on the same concept, and the presence of multiple concepts per image complicates the retrieval of captions that accurately reflect all concepts in a sample.

In the CUB dataset, CLIP-attn continues to dominate SCS-Recall and approaches LLM performance in terms of precision and F1. FALCON* and GRAD-ECLIP struggle with multiple concepts per image and fail to detect uncertainty-related terms, although this does not directly affect SCS metrics. Some captions lack full ground truth concept coverage, penalizing methods like FALCON* that rely heavily on caption content for image cropping and concept extraction.

CELF fine-tuning yields substantial improvements in SCS-precision and F1 compared to CLIP-attn. Training our model with LLM-generated concepts allows it to focus on relevant concepts while reducing noise. For a detailed breakdown of ablation studies evaluating the contributions of different components in the CELF framework for CE, see Appendix G.2. Its performance is also validated against the human study (Appendix G.5).

Table 2: Evaluation of CELF and baselines on CE across datasets across five seeds. Best results in bold, second-best underlined. Metrics: BERTScore recall (BS R.), SCS with Precision (P), Recall (R), and F1.

| **Dataset** | **Metric (%)** | Ablation LLM | Fine-tuned CELF | CLIP-attn | Off-the-shelf FALCON | GRAD-ECLIP |
|---|---|---|---|---|---|---|
| CCM | SCS-R | $80.3 \pm 0.0$ | $82.5 \pm 0.7$ | $\mathbf{83.6 \pm 0.7}$ | $63.0 \pm 0.7$ | $45.2 \pm 0.0$ |
| | SCS-P | $\mathbf{76.1 \pm 0.0}$ | $\underline{72.0 \pm 0.6}$ | $58.6 \pm 0.0$ | $35.7 \pm 0.3$ | $29.9 \pm 0.0$ |
| | SCS-F1 | $\mathbf{78.1 \pm 0.0}$ | $\underline{76.9 \pm 0.6}$ | $68.9 \pm 0.2$ | $45.6 \pm 0.4$ | $40.0 \pm 0.0$ |
| | BS-R | $95.5 \pm 0.0$ | $95.8 \pm 0.1$ | $\underline{95.9 \pm 0.0}$ | $\mathbf{96.5 \pm 0.0}$ | $94.8 \pm 0.0$ |
| VG | SCS-R | $64.7 \pm 0.0$ | $\mathbf{66.0 \pm 0.2}$ | $\underline{65.6 \pm 0.1}$ | $24.8 \pm 0.2$ | $58.1 \pm 0.0$ |
| | SCS-P | $\mathbf{61.6 \pm 0.0}$ | $\underline{59.2 \pm 0.3}$ | $51.2 \pm 0.1$ | $17.1 \pm 0.1$ | $46.7 \pm 0.0$ |
| | SCS-F1 | $\mathbf{63.1 \pm 0.0}$ | $\underline{62.4 \pm 0.2}$ | $57.5 \pm 0.1$ | $20.2 \pm 0.1$ | $51.8 \pm 0.0$ |
| | BS-R | $\underline{94.6 \pm 0.0}$ | $\mathbf{95.4 \pm 0.1}$ | $\mathbf{95.4 \pm 0.0}$ | $92.7 \pm 0.0$ | $93.0 \pm 0.0$ |
| CUB200 | SCS-R | $76.8 \pm 0.0$ | $\mathbf{82.4 \pm 3.0}$ | $\underline{77.1 \pm 0.3}$ | $27.2 \pm 0.2$ | $60.9 \pm 0.0$ |
| | SCS-P | $\underline{84.6 \pm 0.0}$ | $\mathbf{87.0 \pm 1.0}$ | $81.0 \pm 0.1$ | $8.0 \pm 0.1$ | $53.0 \pm 0.0$ |
| | SCS-F1 | $\underline{80.5 \pm 0.0}$ | $\mathbf{84.6 \pm 2.0}$ | $79.0 \pm 0.2$ | $12.4 \pm 0.1$ | $56.7 \pm 0.0$ |
| | BS-R | $95.5 \pm 0.0$ | $\mathbf{96.2 \pm 0.3}$ | $\underline{96.0 \pm 0.1}$ | $92.6 \pm 0.0$ | $94.0 \pm 0.0$ |

## 6.2 CONCEPT RETRIEVAL

In our ablation studies, we evaluate the impact of various components within the CELF framework (Table 3). Both HDBSCAN and hierarchical clustering contribute notably to the performance, with HDBSCAN helping to remove noise and hierarchical clustering further improving concept quality. Additionally, ICC aligns fine-grained concepts and suffices when using well-separated ground truth concepts, while IGC primarily addresses ambiguities introduced by clustering, helping resolve minor confusions. The full CELF model effectively balances these effects, achieving a performance closer to the upper bound observed when using ground truth concepts, which serves as an ideal benchmark. Using a frozen CLIP backbone, we observe that while recall improves, the drop in precision and F1 exceeds 5 $p.p.$ compared to CELF (seed 0), consistent with the fact that CLIP was originally trained for image-caption alignment rather than concept-image alignment, and C-MNIST differs from its pretraining data. Overall, the framework is robust, but the clustering step is seed-dependent, which can introduce variability, as reflected in the standard deviations reported for CELF across seeds. Additional ablation studies exploring the effect of hierarchical clustering thresholds are provided in Appendix G.4.

Table 3: CR results on the complex C-MNIST dataset using seed 0 for all ablation studies and five seeds for CELF. Performance metrics include macro recall, precision, F1-score, and AUC. Optimal thresholds were applied for the multi-label setting. The ablation studies highlight the impact of various components within the CELF framework, such as HDBSCAN and hierarchical clustering (HC), the roles of image-concept (ICC) and image-graph contrastive (IGC) losses.

| Model Variant | Recall (%) | Precision (%) | F1-score (%) | AUC (%) |
|---|---|---|---|---|
| CLIP + Linear Head (Baseline) | 55.5 | 19.1 | 25.8 | 66.7 |
| CELF w/o HDBSCAN | 58.5 | 18.4 | 21.8 | **87.4** |
| CELF w/o HC | 52.9 | 19.7 | 22.6 | 84.2 |
| CELF w/ Frozen Backbone | **63.4** | 33.0 | 39.8 | 81.5 |
| CELF w/ ICC Loss Only | 61.1 | 41.9 | 46.5 | 86.1 |
| CELF w/ IGC Loss Only | 63.1 | **42.4** | **47.4** | 86.3 |
| **CELF (seed 0)** | 62.2 | 42.0 | 47.0 | 86.0 |
| **CELF** | 62.0 ± 3.7 | 39.1 ± 2.4 | 44.1 ± 2.4 | 85.0 ± 1.7 |
| CELF w/ Ground Truth Concepts | 81.3 | 82.3 | 80.5 | 95.5 |

## 6.3 DOWNSTREAM TASK

The results of the downstream task (Table 4) show that while the performance of the ResNet model with retrieved concepts is comparable to the baseline across all metrics, the model with extracted concepts consistently outperforms both the baseline and retrieved concepts in all evaluated metrics. This demonstrates the enhanced utility of extracted concepts in downstream tasks, with the extracted concepts serving as an upper bound for ideal CR. XCB's evaluation has primarily focused on multi-class settings, whereas our experiments involve multi-label classification, potentially limiting XCB's applicability. In addition, it requires a large number of hyperparameters, complicating optimal tuning; our experiments only tested a single configuration. We also assessed VLG-CBM and observed that, although it outperformed XCB, it struggled with concept generation because it relies on LLM prior knowledge rather than dataset context, often producing concepts unrelated to the labels (e.g., "book" for "translation left"). We report additional experiments without image features in Appendix G.7.

## 7 DISCUSSION

CELF broadens the applicability of concept-based interpretability by extracting human-interpretable concepts directly from image captions, eliminating the need for manually annotated concept sets. Unlike traditional CBMs, which rely on class-conditioned concepts, CELF constructs concepts that reflect the semantic content of the data itself. Additionally, CBMs can achieve good performance even when the bottleneck concepts are random or unrelated, limiting their utility for benchmarking concept extraction. Instead, CELF evaluates extracted concepts against caption-derived ground truth, with downstream task evaluation addressed separately in Section 6.3.

Table 4: Performance of ResNet on the CCM dataset (macro-averaged Recall, Precision, F1-Score, and AUC) baseline, with retrieved (RC) and extracted concepts (EC) compared to XCB and VLG-CBM.

| Dataset | Metric (%) | ResNet (Baseline) | ResNet w/ RC | ResNet w/ EC | XCB | VLG-CBM (seed 0) |
|---------|-----------|-------------------|--------------|--------------|-----|------------------|
| CCM | Recall | $94.9 \pm 0.2$ | $94.9 \pm 0.2$ | $\mathbf{95.2 \pm 0.4}$ | $49.2 \pm 2.6$ | 88.9 |
|  | Precision | $91.1 \pm 0.6$ | $91.5 \pm 0.3$ | $\mathbf{92.5 \pm 1.4}$ | $57.4 \pm 7.0$ | 78.9 |
|  | F1-Score | $92.6 \pm 0.3$ | $92.9 \pm 0.2$ | $\mathbf{93.6 \pm 0.9}$ | $45.9 \pm 2.6$ | 82.7 |
|  | AUC | $96.3 \pm 0.1$ | $96.3 \pm 0.1$ | $\mathbf{97.2 \pm 1.3}$ | $85.4 \pm 1.2$ | 92.2 |

Caption-level evaluation has inherent limitations, since captions usually describe only part of the scene. For instance, on CUB, an image-level extractor may correctly identify background elements that are simply absent from the caption. Without manual image annotations, it is unclear whether a "missing" concept is an error or a caption omission. Therefore, we evaluate models on recovering caption-relevant concepts, which best match the available ground truth concepts.

A central challenge in caption-based concept extraction is the variability and noisiness of natural language, which can lead to inconsistent or spurious candidate concepts from LLMs (e.g., prompt leakage rather than the actual caption). CELF addresses this by grounding candidate concepts in captions, removing high-frequency semantically uninformative words, and stabilizing clustering. This yields more stable, dataset-consistent concepts, even if CELF does not always outperform LLMs. Our experiments show that CELF, even with frozen CLIP, surpasses other baselines and remains computationally efficient, while fine-tuning can optionally be applied to further improve precision in settings where domain adaptation is expected to benefit concept quality. Additionally, our ablations (see Table 11) show that attention-based extraction with pretrained CLIP substantially reduces noise relative to similarity-based extraction, and the human study in Appendix G.5 confirms that CELF's extracted concepts align with human annotations.

For CR, quality depends on the underlying VLM. Concept extraction does not require perfectly aligned image-text pairs, but retrieval performance is constrained by domain shift and pretraining biases. This suggests that pairing CELF with domain-specialized VLMs is a natural next step, particularly in safety-critical areas such as clinical imaging, where captions are noisy and visual semantics differ substantially from general-purpose datasets.

CELF's modular design also allows CE and CR to be used independently. As CE relies on captions, it cannot be applied directly to image-only benchmarks such as CIFAR, but CR can still be applied by retrieving concepts extracted from a separate captioned dataset and pairing them with images.

## 8 CONCLUSION, POTENTIAL LIMITATIONS AND FUTURE WORK

We introduced CELF, a self-supervised, multimodal framework for extracting and retrieving human-interpretable concepts without manual annotations. CELF mitigates key issues in concept-based XAI, including CLIP noise sensitivity, and outperforms gradient-based methods across multiple datasets, demonstrating strong generalization. We also proposed C-MNIST, a configurable dataset for controlled benchmarking, and introduced SCS, a conservative semantic metric for evaluating concept extraction. CELF's consistent gains in both synthetic and real-world settings, including downstream tasks, highlight its scalability and practical utility in low annotation settings.

However, CELF requires image-caption pairs for CE, which limits its applicability in domains lacking aligned text. Although it improves over CLIP, some extracted words remain noisy. Additionally, while SCS is effective, it could benefit from synonym-aware tuning. In future work, we aim to: (1) extend CELF to domain-specialized settings (e.g. clinical); (2) investigate hierarchical architectures for better concept grouping; (3) explore pretraining objectives that improve focus on semantically relevant image regions; (4) evaluate CELF on other SOTA VLMs; and (5) assess concept generalization on attribute-substitution benchmarks such as SUB. More broadly, CELF's self-supervised objective offers the potential to learn domain-general "foundational concepts" when scaled to large, diverse image-text corpora, which could substantially expand the scope of concept-based interpretability.

## 9 REPRODUCIBILITY STATEMENT

To support reproducibility, we plan to publicly release the code for both CELF and C-MNIST. C-MNIST has already been approved for open-source release, while CELF is currently under evaluation for approval. The released version of C-MNIST will include improvements compared to the version used in this paper. Nevertheless, the Appendix (Sections B, C, and F) provides detailed descriptions of the training procedures, model configurations, and loss functions, ensuring that all experiments can be replicated under equivalent conditions.

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

# Appendix

## Table of Contents

## A  ADDITIONAL RELATED WORKS

Concept-based XAI has recently gained significant attention, particularly approaches that aim to avoid relying on annotated concepts. In this section, we review methods that share this motivation but do not align with our CE setting; for this reason, they are not discussed in the main body. Table 5 summarizes their key properties and limitations, except CRP (Achtibat et al., 2023), which serves a different purpose, and the work of Kim et al. (2023), which is applied in the medical domain.

Methods such as LaBo (Yang et al., 2023), CLBM (Yan et al., 2023a) and LF-CBM (Oikarinen et al., 2023) generate class-conditioned concept sets using LLMs and subsequently train CBMs. These methods depend on the LLM's internal knowledge to propose concepts, which can lead to hallucinations or omissions of class-relevant attributes. In contrast, CELF uses an LLM only as an instruction-following tool to extract concepts that are explicitly grounded in captions, rather than relying on the LLM's prior knowledge to infer concepts.

Some works (Kim et al., 2023) explore CBMs in domains such as medical imaging, where concepts are generated for each class and visually relevant concepts are selected via submodular optimization. This highlights the potential for domain-specific adaptations, and we plan to extend CELF to medical and other specialized domains where captions or textual descriptions may be noisy or sparse.

SpLiCE (Bhalla et al., 2024) extracts frequent words or short phrases from captions, removes high similar concepts to reduce redundancy, and selects a fixed number of concepts via a top-K procedure. This introduces two limitations: the concept length is fixed and the top-K filtering may select non-semantic tokens or miss semantically important ones. CELF, instead, aims to recover all relevant concepts in a dataset without imposing a fixed concept length or vocabulary size.

V2C-CBM (He et al., 2025) relies on a predefined concept vocabulary and therefore cannot discover new concepts that emerge from the data. Text-to-Concept (Moayeri et al., 2023) embeds provided textual strings as concept vectors and aligns them with images, without generating textual concepts. By contrast, CELF automatically extracts candidate concepts directly from captions. Finally, CRP (Achtibat et al., 2023) focuses on propagating relevance backward in an image classification model into visual regions and manually associating them with textual concepts, serving a different goal than our automated concept extraction pipeline.

## B  CONCEPTUAL-MNIST: ADDITIONAL DETAILS

### B.1  DIGIT ROTATION DETECTION

For digit orientation alignment, three specific rotation methods were implemented:

1. For digits with distinct vertical or horizontal lines, this method corrects the orientation by aligning those lines to the intended axis.

2. For digits "1" (with only a vertical line) and "0," a simple rectangle fitting is sufficient for alignment.

3. Used for all digits except "5," this method aligns each digit with a reference image of correctly oriented digits. Images are binarized and skeletonized to reduce noise, and a feature detection and description algorithm identifies keypoints for alignment. To ensure accuracy, Lowe's ratio test filters matches, and the affine transformation matrix allows for the rotation calculation.

The optimal method is selected based on a confidence level, defined as the similarity between the rotated and reference images. If the confidence level is less or equal to an empirically determined threshold of 94%, the sample is excluded from the dataset. This threshold was selected based on observed confidence levels in both successful and failed attempts at detecting rotation.

### B.2  PROMPT FOR DESCRIPTION GENERATION

system: Generate a very simple description about an image of a digit (also called object) with the given characteristics by the user. For example, the user gives green, high saturation, low

Table 5: Comparison of CELF with other CE and CR methods, highlighting key differences and limitations. $\sim$ indicates partial CE, as LaBo first generates descriptions and then extracts concepts and SpLiCE extracts one and two word n-grams to form the concept set.

| Method | CE | CR | Strengths | Limitations |
|---|---|---|---|---|
| LaBo (Yang et al., 2023) | $\sim$ | ✓ | LLM-guided generation of class attributes; lightweight training for CBM construction | Relies on LLM-generated candidate concepts and CLIP alignment; susceptible to LLM hallucination |
| CLBM (Yan et al., 2023a) | × | ✓ | Queries LLMs to generate a large attribute pool, then prunes to a concise subset via learning-to-search | Quality depends on the LLM-generated attribute pool |
| LF-CBM (Oikarinen et al., 2023) | × | ✓ | Refines class-wise noisy concepts using filtering | Susceptible to VLM noise propagation |
| SpLiCE (Bhalla et al., 2024) | $\sim$ | ✓ | Produces sparse linear concept representations inside CLIP; no additional training required | Concept vocabulary derived from frequent 1–2 word n-grams in LAION; may miss domain-specific concepts |
| V2C-CBM (He et al., 2025) | × | ✓ | Builds an image-driven concept codebook and filters candidates with VLM similarity (no LLM required) | Adopts a heuristically constructed vocabulary (common words, n-grams); may miss domain-specific concepts |
| Text-to-Concept (Moayeri et al., 2023) | × | ✓ | Learns cross-model alignment so that text embeddings act as concept vectors | Requires provided text strings; quality depends on VLM embeddings |

> decrease in size and high translation to left so you must generate something like: The image has a digit of blue color, high saturation, translated to the left and small.

> user: Generate a simple description of an image featuring an {type_obj} that has the following list of characteristics on the {type_obj} itself: {digit_transf} and the following on the background {background_transf}.

## C  DATASETS ADDITIONAL INFORMATION

### C.1  SYNTHETIC DATASETS

In this section, examples of the components in the complex C-MNIST are shown in Table 6. Table 8 displays the distribution of concepts.

**Concept Label Mapping.**    To reduce label sparsity and improve generalization, fine-grained concept annotations in C-MNIST were grouped into coarser, semantically meaningful labels. For example, specific rotation operations like *'second quadrant rotation'* and *'vertical flip'* were merged under the broader concept *'rotated'*. This mapping enables more robust evaluation of concept extraction while reducing the number of output classes. The full mapping includes:

- **translation left** $\rightarrow$ *minor translation to left*
- **different position** $\rightarrow$ *both horizontal and vertical flip, horizontal flip, quadrant rotations, vertical flip*

Table 6: Examples of complex C-MNIST, including images, descriptions, and concepts.

| Image | Description | Concepts |
|---|---|---|
|  | The image shows an digit that has undergone both horizontal and vertical flips, has a medium size, and moderate thickness. The background is featureless. | both horizontal and vertical flip, medium size, moderate thickness |
|  | The image shows an digit that has undergone a horizontal flip, is of minor thickness, and has a small size. The background has a spring-like appearance, is significantly saturated, and has a moderate level of brightness. | horizontal flip, minor thickness, small size, spring, significant saturation, moderate brightness |
|  | The image shows an square with a red color, significant saturation, moderate thickness, and a medium size. The square has a significant brightness and a stone texture. The background is empty. | moderate thickness, medium size, red, significant saturation, significant brightness, stone texture |

- **rotated** → *first to fourth quadrant rotations*
- **lower scale** → *small size*
- **texture** → *wood, tile, stone textures*
- **rgb** → *red, green, blue*
- **bright** → *moderate and significant brightness*
- **low saturation** → *minor saturation*
- **thick** → *moderate and significant thickness*
- **off-center** → *various translations to left or right*

This abstraction helps balance the dataset and emphasizes semantically meaningful groupings of visual transformations and styles.

## C.2 VISUAL GENOME

The Visual Genome dataset comprises over 108,000 images with detailed annotations (Table 9) (Krishna et al., 2016). Each image is labeled with multiple objects, attributes, and relationships, providing a rich foundation for concept extraction. The dataset includes:

- Object annotations: Bounding boxes with corresponding object labels.
- Attribute annotations: Descriptions of object properties (e.g., "red car," "wooden table").
- Relationship annotations: Pairwise relationships between objects (e.g.,"man sitting on chair").

To construct caption–concept pairs for training and evaluation, we leverage the region-level descriptions associated with annotated bounding boxes. While Visual Genome often provides multiple descriptions for the same region, many of which refer to the same object or concept, we aim to increase the semantic diversity of the resulting captions. To achieve this, we encode the region descriptions using a Sentence-Transformers model (Reimers & Gurevych, 2019) and compute pairwise similarities. From these, we select a subset of dissimilar descriptions to ensure coverage of varied linguistic expressions and conceptual aspects. These are then merged into a single caption that

Table 7: Frequency of concepts in train, validation, and test data of complex Conceptual-MNIST.

| Concepts | Train | Validation | Test |
|---|---|---|---|
| moderate thickness | 50,450 (75.3%) | 10,268 (76.2%) | 8,766 (77.9%) |
| minor translation to left | 41,576 (62.1%) | 7,887 (58.5%) | 5,803 (51.6%) |
| small size | 27,728 (41.4%) | 5,577 (41.4%) | 4,549 (40.4%) |
| medium size | 19,034 (28.4%) | 3,924 (29.1%) | 3,241 28.8%) |
| minor thickness | 16,508 (24.6%) | 3,203 (23.8%) | 2,479 (22.0%) |
| tiles texture | 9,328 (13.9%) | 1,941 (14.4%) | 1,565 (13.9%) |
| minor staturation | 9,327 (13.9%) | 2,541 (18.8%) | 1,529 (13.6%) |
| significant brightness | 9,309 (13.9%) | 1,845 (13.7%) | 1,572 (14.0%) |
| moderate brightness | 9,318 (13.9%) | 1,820 (13.5%) | 1,568 (13.9%) |
| wood texture | 9,249 (13.8%) | 1,817 (13.5%) | 1,609 (14.3%) |
| significant saturation | 9,231 (13.8%) | 1,877 (13.9%) | 1,542 (13.7%) |
| moderate saturation | 9,259 (13.8%) | 1,859 (13.8%) | 1,591 (14.1%) |
| minor brightness | 9,190 (13.7%) | 1,871 (13.9%) | 1,522 (13.5%) |
| stone texture | 9,187 (13.7%) | 1,900 (14.1%) | 1,572 (14.0%) |
| horizontal flip | 5,610 (8.4%) | 1,168 (8.7%) | 914 (8.1%) |
| fourth quadrant rotation | 7,001 (105%) | 1,405 (10.4%) | 1,177 (10.5%) |
| second quadrant rotation | 7,023 (10.5%) | 1,452 (10.8%) | 1,179 (10.5%) |
| third quadrant rotation | 6,935 (10.4%) | 1,411 (10.5%) | 1,191 (10.6%) |
| first quadrant rotation | 6,904 (10.3%) | 1,391 (10.3%) | 1,149 (10.2%) |
| vertical flip | 4,347 (6.5%) | 858 (6.4%) | 742 (6.6%) |
| blue | 2,429 (3.6%) | 467 (3.5%) | 412 (3.7%) |
| rose | 2,405 (3.6%) | 450 (3.3%) | 404 (3.6%) |
| azure | 2,402 (3.6%) | 464 (3.4%) | 379 (3.4%) |
| violet | 2,328 (3.5%) | 490 (3.6%) | 396 (3.5%) |
| green | 2,326 (3.5%) | 431 (3.2%) | 370 (3.3%) |
| orange | 2,270 (3.4%) | 445 (3.3%) | 402 (3.6%) |
| red | 2,295 (3.4%) | 499 (3.7%) | 381 (3.4%) |
| yellow | 2,240 (3.3%) | 464 (3.4%) | 388 (3.4%) |
| spring | 2,274 (3.4%) | 468 (3.5%) | 418 (3.7%) |
| magenta | 2,286 (3.4%) | 480 (3.6%) | 354 (3.1%) |
| chartreuse | 2,283 (3.4%) | 467 (3.5%) | 377 (3.3%) |
| cyan | 2,279 (3.4%) | 411 (3.0%) | 381 (3.4%) |
| both horizontal and vertical flip | 920 (1.4%) | 172 (1.3%) | 126 (1.1%) |
| minor translation to right | 241 (0.4%) | 534 (4.0%) | 1,294 (11.5%) |
| moderate translation to right | 14 (0.02%) | 10 (0.1%) | 21 (0.2%) |
| significant translation to right | 2 (0.00%) | - | 1 (0.01%) |
| significant thickness | 37 (0.06%) | 11 (0.1%) | 10 (0.1%) |

Table 8: Frequency of labels in train, validation, and test data of complex Conceptual-MNIST.

| Labels | Train | Validation | Test |
|---|---|---|---|
| thick | 50,453 (75.4%) | 10,270 (76.2%) | 8,763 (78.0%) |
| different position | 34,818 (52.0%) | 7,025 (52.1%) | 5,858 (52.1%) |
| off-center | 41,814 (62.5%) | 8,428 (62.6%) | 7,109 (63.3%) |
| lower scale | 27,711 (41.4%) | 5,573 (41.4%) | 4,539 (40.4%) |
| texture | 18,427 (27.5%) | 3,713 (27.6%) | 3,173 (28.2%) |
| rgb | 7,045 (10.5%) | 1,395 (10.4%) | 1,163 (10.3%) |
| translation left | 41,557 (62.1%) | 7,884 (58.5%) | 5,795 (51.6%) |
| rotated | 27,851 (41.6%) | 5,655 (42.0%) | 4,693 (41.8%) |
| bright | 18,612 (27.8%) | 3,663 (27.2%) | 3,135 (27.9%) |
| low saturation | 9,320 (13.9%) | 1,798 (13.3%) | 1,525 (13.6%) |

Table 9: Overview of the information in the Visual Genome dataset, including object, attribute, and relationship classes. The information was retrieved from (Krishna et al., 2016).

| | |
|---|---|
| **Number of Images** | 108,000 |
| **Descriptions per Image** | 50 |
| **Total Objects** | 4,102,818 |
| **Number of Objects Categories** | 76,340 |
| **Objects per Image** | 16 |
| **Number of Attributes Categories** | 15,626 |
| **Attributes per Image** | 16 |
| **Number of Relationships Categories** | 47 |
| **Relationships per Image** | 18 |
| **Questions Answers** | 1,773,258 |

contains rich and diverse language, and paired with the corresponding object attributes as ground truth concepts. This process is repeated to create multiple diverse caption–concept pairs per image.

## C.3   CUB

The CUB-200-2011 dataset contains 11,788 images across 200 bird species with fine-grained annotations (Alukaev et al., 2023). It is widely used for fine-grained classification and visual attribute prediction. Each image is annotated with:

- Class labels: One of 200 bird species, providing fine-grained categorical supervision.
- Part annotations: Locations of 15 predefined body parts (e.g.,"beak," "left wing") with (x, y) coordinates.
- Attribute annotations: 312 binary attributes describing color, shape, size, and patterns (e.g.,"has blue wings," "has striped belly").

## D  PROMPT FOR PSEUDO-LABEL GENERATION

The prompt for pseudo-label generation was inspired in the following prompt from (Shi et al., 2023):

> You are a knowledge graph extractor, and your task is to extract and return a knowledge graph from a given text.Let's extract it step by step: (1). Identify the entities in the text. An entity can be a noun or a noun phrase that refers to a real-world object or an abstract concept. You can use a named entity recognition (NER) tool or a part-of -speech (POS) tagger to identify the entities. (2). Identify the relationships between the entities. A relationship can be a verb or a prepositional phrase that connects two entities. You can use dependency parsing to identify the relationships. (3). Summarize each entity and relation as short as possible and remove any stop words. (4). Only return the knowledge graph in the triplet format: ('head entity', 'relation ', 'tail entity'). (5). Most importantly, if you cannot find any knowledge, please just output:"None". Here is the content: [x]

The prompt was fine-tuned according to the observed generated text, with the final chat history given to the LLM for complex C-MNIST:

> user: You are a knowledge graph extractor, and your task is to extract and return a knowledge graph from a given text.Let's extract it step by step: (1). Identify the entities in the text. An entity can be a noun or a noun phrase that refers to a real-world object or an abstract concept. You can use a named entity recognition (NER) tool or a part-of -speech (POS) tagger to identify the entities. (2). Identify the relationships between the entities. A relationship can be a verb that connects two entities that should be present in the content. You can use dependency parsing to identify the relationships. (3). Summarize each entity and relation as short as possible and remove any stop words. (4). Only return the knowledge graph in the triplet format: (head entity, relation , tail entity). (5). Most importantly, if you cannot find any knowledge, please just output:"None". Here is the content: The image shows an object with green color against a background with a very high saturation.

> assistant:(object, has, green color);(background, has, very high saturation)

> user: You are a knowledge graph extractor, and your task is to extract and return a knowledge graph from a given text.Let's extract it step by step: (1). Identify the entities in the text. An entity can be a noun or a noun phrase that refers to a real-world object or an abstract concept. You can use a named entity recognition (NER) tool or a part-of -speech (POS) tagger to identify the entities. (2). Identify the relationships between the entities. A relationship can be a verb that connects two entities that should be present in the content. You can use dependency parsing to identify the relationships. (3). Summarize each entity and relation as short as possible and remove any stop words. (4). Only return the knowledge graph in the triplet format: (head entity, relation , tail entity). (5). Most importantly, if you cannot find any knowledge, please just output:"None". Here is the content:{descr}

## E  LANGUAGE MODEL CHOICES AND JUSTIFICATIONS

For reproducibility and transparency, we detail the reasoning behind our language model selections across different stages of the pipeline:

- **Caption generation**: We used an open-source model with strong performance. The choice was pragmatic rather than driven by specific requirements.
- **Pseudo-label generation**: Zephyr was initially used due to its strong reasoning capabilities, but it proved computationally expensive. We opted for Phi-3 for these experiments due to its significantly faster generation speed while maintaining reasonable performance.
- **Clustering (encoding)**: Various sentence encoders were tested. Since this step requires only text embeddings and not generation, using a full LLM was unnecessary. We selected `paraphrase-MiniLM-L6-v2` from Sentence-Transformers (Reimers & Gurevych, 2019) due to its efficiency and robustness to word order, an important feature given CELF's sensitivity to this.

- **SCS**: To avoid evaluation bias, we used a model different from the one used during clustering. This ensures that semantic similarity is assessed independently of the encoding model used earlier in the pipeline.

# F  METHODS: TRAINING SETUP

## F.1  CONCEPT EXTRACTION

The experiment was conducted with a batch size of 64, we employed the Adam optimizer (Kingma & Ba, 2014) with an initial learning rate of $1 \times 10^{-5}$, in conjunction with a cosine annealing scheduler (Loshchilov & Hutter, 2017) that had a maximum of 50 iterations and a minimum learning rate of $1 \times 10^{-7}$. Early stopping was implemented based on the evolution of the validation loss and F1-Score. The parameter $\beta$ was set to 3 for the first 9 epochs and reduced to 1.5 for the remaining epoch, while $\alpha$ was fixed at 1, as the loss associated with attention weights directly aligned with the objective.

Furthermore, registers were incorporated in accordance with the findings presented in (Darcet et al., 2024), with the objective of reducing noise within the ViT.

## F.2  CONCEPT CLUSTERING

We embedded concepts with the paraphrase-MiniLM-L6-v2 model from Sentence-Transformers (Reimers & Gurevych, 2019), which performs well at capturing semantic similarity, including variations in word order.

We apply UMAP for dimensionality reduction to 10 dimensions, as in (Allaoui et al., 2020), to retain coarse-grained semantic structure while reducing noise. Clustering is then performed using Hierarchical Density-Based Spatial Clustering of Applications with Noise (HDBSCAN) (McInnes et al., 2017). To address low-confidence clusters (confidence $< 50\%$), we perform a second clustering stage in the original 384-dimensional space. This allows finer-grained distinctions to be captured. The clustering was configured with the following HDBSCAN hyperparameters: min_cluster_size=2, cluster_selection_method='leaf', allow_single_cluster=True, and approx_min_span_tree=False using the HDBSCAN implementation from scikit-learn (Pedregosa et al., 2011). A minimum cluster size of 2 concepts and the leaf-type cluster selection method were used to obtain the finest and most homogeneous clusters.

However, as semantically equivalent concepts sometimes remained split across clusters, we merged clusters whose centroid cosine similarity exceeded 90%. This threshold was empirically selected based on concept distribution in the C-MNIST train set clusters. Clusters with lower similarity were retained separately but associated with broader parent clusters to preserve contextual relevance.

## F.3  CONCEPT RETRIEVAL

The image–concept loss generalizes the contrastive principle to a multi-label setting using binary cross-entropy, enabling fine-grained supervision at the concept level. Specifically:

- **Image–Graph Contrastive Loss:** This objective aligns image embeddings with their corresponding concept graph embeddings. Following CLIP setup, we apply a symmetric cross-entropy loss over cosine similarity scores, scaled by a learnable temperature parameter:

$$\mathcal{L}_{\text{img-graph}} = \frac{1}{2}\left(\text{CE}(\text{sim}(v_i, g_j), i) + \text{CE}(\text{sim}(g_i, v_j), i)\right),$$

  where $v_i$ and $g_i$ denote the image and concept graph embeddings, respectively. This formulation closely resembles the original InfoNCE loss used in CLIP, applied at the batch level.

- **Image–Concept Contrastive Loss:** To promote fine-grained alignment between image embeddings and individual concept representations, we introduce a multi-label contrastive loss. Given a binary relevance matrix $M_{ij}$ between image $v_i$ and concept $c_j$, we compute:

$$\mathcal{L}_{\text{img-concepts}} = \text{BCEWithLogits}(\text{sim}(v_i, c_j), M_{ij}),$$

where the binary cross-entropy loss reflects the multi-label nature of the concept prediction task.

Both components share a common logit scaling factor (learned during training), and the final loss is given by:

$$\mathcal{L} = \alpha \mathcal{L}_{\text{img-graph}} + \beta \mathcal{L}_{\text{img-concepts}}.$$

with $\alpha = 0.25$ and $\beta = 1.0$.

We fine-tune the CLIP backbone using the Adam optimizer, applying a learning rate of $1 \times 10^{-5}$ for the pretrained CLIP parameters and a weight decay of $1 \times 10^{-4}$. We use a batch size of 32. The learning rate is scheduled with cosine annealing and a linear warm-up over the first 5% of training steps, based on the recommendations in (Gupta et al., 2023). Early stopping with a patience of 5 epochs is employed if the validation mean Average Precision does not improve to prevent overfitting.

In the second stage, we freeze the parameters of the fine-tuned CLIP model and train the new layers. The remaining model parameters are optimized using the Adam optimizer with the following settings: a learning rate of $1 \times 10^{-4}$ and a weight decay of $1 \times 10^{-4}$. The learning rate follows a cosine annealing schedule with a minimum learning rate of $1 \times 10^{-6}$ over all epochs. The batch size for training is set to 32. Early stopping with a patience of 10 epochs is employed if the validation AUC does not improve.

### F.4 MULTI-LABEL CLASSIFICATION TASK

We set the batch size to 64 for training. The model parameters are optimized using the Adam optimizer with a learning rate of $1 \times 10^{-4}$. The learning rate is adjusted using the `ReduceLROnPlateau` scheduler, which reduces the learning rate by a factor of 0.5 when the validation AUC has not improved for 2 consecutive epochs. The training loop is designed to run for a maximum of 50 epochs. We use early stopping with a patience value of 10 epochs if the validation AUC does not improve during this period.

**XCB Training Setup.** For the XCB baseline, we follow a concept bottleneck architecture with a pretrained ResNet-50 backbone as the feature extractor, followed by a concept extractor based on an attention mechanism with positional encodings and slot normalization. The concept extractor maps inputs to a vocabulary of 770 concepts using a Gumbel-Sigmoid activation, chosen to match the vocabulary size of the complex C-MNIST dataset. The predictor module is a two-layer MLP that maps the 40-dimensional concept representation to 10 output labels. We use binary cross-entropy as the main task loss and Jensen-Shannon divergence as the tie loss, weighted by a factor of 10. The model is optimized using Adam (learning rate $1 \times 10^{-4}$) for the predictor and AdamW (learning rate $4 \times 10^{-3}$, weight decay 0.03) for the concept extractor, each with its own learning rate scheduler: `ReduceLROnPlateau` for the predictor and `MultiStepLR` for the concept extractor.

**VLG-CBM Training Setup.** We generate concepts using GPT3.5 (`gpt-3.5-turbo-instruct`) and a 0.15 text threshold, as higher values failed to capture concepts for images. For the backbone, we use the pretrained CLIP-RN50 model. Most thresholds and configuration parameters follow those used for the CUB dataset, except for the batch size, learning rate, weight decay, and number of concept-bottleneck training epochs, which match our own setup. Finally, we adapt the training of the sparse final layer for a multi-label setting.

## G ADDITIONAL RESULTS

### G.1 SCS VALIDATION

To further validate SCS, we conducted a perturbation study (from approximately 70 to 100 samples) using C-MNIST concepts. We generated perturbed concept pairs for each of the following categories until all feasible substitutions were covered:

- Synonym substitution: Replacing one word in a concept with its synonym (e.g., replacing "minor" with "low" or "small").

- Antonym substitution: Replacing a word with its opposite (e.g., "left", "right").

- Unrelated concepts: Comparing concepts from CUB200 with those from C-MNIST to simulate dissimilar or out-of-domain contexts.

SCS was compared with BLEU and BERTScore in Table 10. BLEU focuses on exact word overlap and is unable to capture semantic similarity, which results in low scores even for synonymous concepts. In contrast, BERTScore tends to overestimate similarity, assigning high scores even to clearly unrelated concepts (e.g., "significant saturation" vs. "probably buff nape").

Table 10: Evaluation of semantic similarity metrics (BLEU, BERTScore, and SCS) on manually labelled synonym, antonym, and unrelated concept pairs.

|  | Synonym | Antonym | Unrelated |
|---|---|---|---|
| BERTScore | 97.9% | 98.5% | 81.6% |
| BLEU | 23.3% | 21.0% | 0.4% |
| SCS | 74.3% | 53.8% | 0.0% |

SCS strikes a better balance: it is sensitive to semantic differences while still recognising meaningful similarities. It scored 74.3% on synonym pairs (highlighting its ability to capture semantic equivalence), 53.8% on antonyms (showing some sensitivity to semantic opposition), and 0.0% on unrelated pairs (demonstrating strong discrimination). The BLEU score is non-zero for unrelated pairs because both CUB200 and C-MNIST mention colors; C-MNIST mentions only the color, while CUB200 also provides context/localization (e.g., "probably red beak"), which leads to partial word overlap.

To guide threshold selection, we manually inspected similarity scores in C-MNIST. For example: "horizontal vertical flips" vs. "both horizontal and vertical flip": 92.3%, "minor translation" vs. "minor translation left": 76.5%, and "moderate minor saturation brightness" vs. "moderate brightness": 79.0%. These observations suggest that SCS can effectively rank semantic similarity. However, we acknowledge that SCS still misses certain synonymous expressions due to its conservative similarity threshold. This could be improved by fine-tuning the underlying language model using synonym databases (e.g., WordNet or ConceptNet), which we plan to explore in future work.

### G.2 CONCEPT EXTRACTION

#### G.2.1 STATISTICAL ANALYSIS

In addition to per-seed variability, we performed statistical tests to evaluate significance across tasks. CELF consistently outperforms baselines like FALCON and GRAD-ECLIP with higher mean scores across tasks ($mean = 0.7484$, $std = 0.1121$), while the pairwise Wilcoxon tests confirm CELF's advantage over other methods. While the p-values are not below 0.05, likely due to the small number of tasks, the confidence intervals for CELF–FALCON and CELF–GRAD-ECLIP comparisons ($\Delta = +0.4$, $CI = [0.26, 0.64]$ and $\Delta = +0.27$, $CI = [0.11, 0.41]$) suggest consistent improvements.

#### G.2.2 ABLATION STUDIES

To evaluate the contribution of each component in our framework, we compare four configurations: (1) pretrained CLIP with similarity-based concept extractionn, where each candidate word is individually passed through CLIP and its similarity to the image embedding is used as the selection criterion; (2) pretrained CLIP with attention-based extraction; (2) pretrained CLIP with attention-based extraction; (3) CLIP fine-tuned with contrastive loss; and (4) our full CELF model.

As shown in Table 11, CELF achieves the highest performance across all SCS metrics on the CCM dataset, with notable improvements in precision and F1 over both standard and fine-tuned CLIP variants. These results indicate that CELF is more effective at extracting semantically relevant concepts. While fine-tuning improves SCS, it still underperforms compared to CELF, highlighting the importance of the keyphrase extraction task.

Additionally, attention-based extraction outperforms similarity-based retrieval by a large margin. This suggests that focusing on contextual word-to-word relationships, rather than isolated similarity to the image, improves generalization across datasets.

Table 11: Ablation results across five seeds on the CCM dataset. We report SCS Recall (SCS-R), Precision (SCS-P), and F1-score (SCS-F1), along with BERTScore Recall (BS-R). CELF consistently outperforms other variants across SCS metrics. CLIP-sim and CLIP-attn refer to concept extraction using similarity and attention weights, respectively.

| CCM Metrics (%) | CLIP sim | CLIP attn | fine-tuned CLIP | CELF |
|---|---|---|---|---|
| SCS-R | $18.5 \pm 0.0$ | $\mathbf{83.6 \pm 0.7}$ | $81.0 \pm 1.6$ | $\underline{82.5 \pm 0.7}$ |
| SCS-P | $30.3 \pm 0.1$ | $58.6 \pm 0.0$ | $\underline{62.3 \pm 0.6}$ | $\mathbf{72.0 \pm 0.6}$ |
| SCS-F1 | $23.0 \pm 0.1$ | $68.9 \pm 0.2$ | $\underline{70.4 \pm 0.9}$ | $\mathbf{76.9 \pm 0.6}$ |
| BS R | $89.0 \pm 0.1$ | $\mathbf{95.9 \pm 0.0}$ | $\mathbf{95.9 \pm 0.1}$ | $\underline{95.8 \pm 0.1}$ |

Heatmap analysis (Figure 3) further confirms that the fine-tuned CLIP model tends to miss on relevant words, whereas CELF presents high confidence in all of them.

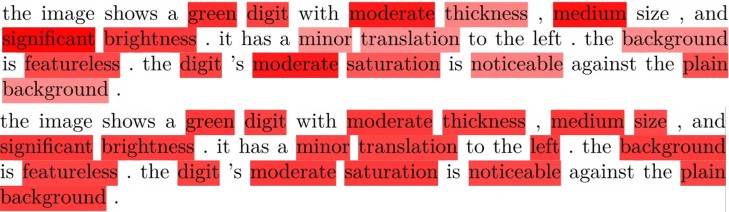

Figure 3: Heatmap of the attention scores for extracted words (scores above 0.4). Intense red indicates higher attention, while lighter red denotes lower attention. The top phrase shows results using the fine-tuned CLIP, while the bottom phrase shows results from CELF. Here, the LLM failed to generate pseudo-labels.

Nonetheless, some limitations persist, particularly when pseudo-labels are incomplete or when multiple concepts share overlapping words. These challenges make accurate concept grouping more difficult (see Figure 4).

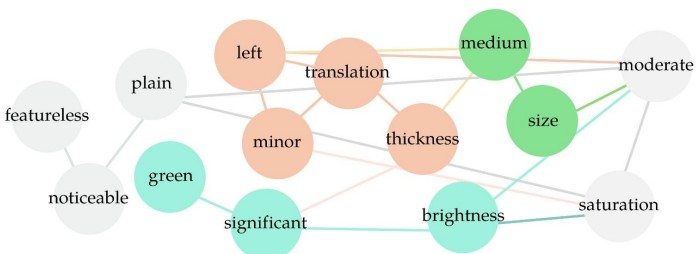

Figure 4: Visual representation of the concept graph generated from CELF's extracted words without filtering localization terms. Different colors indicate word groupings. For clarity, not all links are displayed. In this case, some concepts are mixed together, such as "green" and "significant brightness."

### G.2.3 THRESHOLD SELECTION AND SENSITIVITY ANALYSIS IN WORD FILTERING

We experimented with three thresholds as shown in Table 13. We observe that threshold 0.4 yields the best overall F1 scores for CCM and VG, while CUB performs best with threshold 0.3. This is likely because CUB contains clean, synthetic captions built from templates such as "The bird has {concepts}", making it less noisy and more sensitive to a lower threshold. In contrast, VG and CCM involve greater variability and noise in textual descriptions. In these settings, threshold 0.4 improves

precision substantially while only modestly reducing recall. Thus, it serves as a conservative yet effective choice for balancing noise reduction and concept coverage.

Table 13: Sensitivity study of different word selection thresholds for CELF on CE across five seeds. Best results are shown in bold, and second-best results are underlined. Metrics include BERTScore Recall (BS R.) and SCS with Precision (P), Recall (R), and F1.

| Dataset | Metric (%) | Word selection threshold | | |
|---------|------------|--------------|--------|--------|
| | | 0.3 | 0.4 | 0.5 |
| CCM | SCS-R | $85.4 \pm 0.1$ | $82.5 \pm 0.7$ | $82.0 \pm 1.2$ |
| | SCS-P | $61.9 \pm 0.2$ | $72.0 \pm 0.6$ | $71.9 \pm 0.8$ |
| | SCS-F1 | $71.7 \pm 0.2$ | $76.9 \pm 0.6$ | $76.7 \pm 0.9$ |
| | BS-R | $96.0 \pm 0.1$ | $95.8 \pm 0.1$ | $95.8 \pm 0.1$ |
| CUB200 | SCS-R | $94.4 \pm 0.8$ | $82.6 \pm 2.7$ | $80.3 \pm 2.2$ |
| | SCS-P | $88.8 \pm 0.1$ | $87.1 \pm 0.8$ | $86.4 \pm 0.7$ |
| | SCS-F1 | $91.5 \pm 0.4$ | $84.8 \pm 1.8$ | $83.2 \pm 1.5$ |
| | BS-R | $97.9 \pm 0.1$ | $96.2 \pm 0.3$ | $95.9 \pm 0.3$ |
| VG | SCS-R | $71.2 \pm 0.2$ | $66.0 \pm 0.2$ | $54.2 \pm 0.5$ |
| | SCS-P | $50.9 \pm 0.1$ | $59.2 \pm 0.3$ | $62.5 \pm 0.3$ |
| | SCS-F1 | $59.4 \pm 0.2$ | $62.4 \pm 0.2$ | $58.1 \pm 0.4$ |
| | BS-R | $95.9 \pm 0.1$ | $95.4 \pm 0.1$ | $93.9 \pm 0.1$ |

### G.3 FALCON*: FALCON ADAPTATION

FALCON was not originally designed for image-based concept extraction, so we made several modifications to adapt it. First, we filtered captions to retain only those that contained all the sample concepts. If no such captions were found, we included captions containing at least one sample concept. FALCON* consists of two models: a ResNet-based encoder and CLIP. For consistency with our framework, we replaced the ResNet encoder with a pretrained CLIP model. The concept extraction then proceeded as outlined in the original FALCON* paper. However, as contrastive interpretability was difficult to adapt for image-based tasks, we chose not to apply it here, knowing that this would likely affect the precision score.

For the Visual Genome dataset, some image-text pairs contained no phrases corresponding to any of the target concepts, so we removed them from evaluation, as including them would unfairly penalize the method.

### G.4 CONCEPT RETRIEVAL: CLUSTERING THRESHOLD SENSITIVITY

We conducted a sensitivity analysis on the hierarchical clustering threshold. Results (Table 14) show that increasing the threshold from 0.9 to 0.95 yields minor improvements in F1 and AUC. This is likely due to tighter cluster boundaries that reduce intra-cluster semantic noise, even though the total number of clusters remained constant. However, these differences are small and do not alter our core findings.

However, as expected, lowering the threshold to 0.85 produces the worst results in terms of precision and F1. A more permissive threshold leads to larger, broader clusters, which increases semantic overlap and reduces the discriminative power of the extracted concepts.

### G.5 HUMAN STUDY

To validate our framework, and given that the C-MNIST dataset captions contained slight variations and occasional hallucinations from the language model (e.g., mentioning a rotation of 180°), we conducted a small human study. In this study, participants were asked to annotate the first 50 captions from the test set.

**Annotation protocol.** Participants were instructed to identify two categories of concepts within each caption:

Table 14: Sensitivity study of different thresholds for hierarchical clustering on the complex C-MNIST dataset. Performance metrics include macro Recall, Precision, F1-score, and AUC. Optimal classification thresholds were used for the multi-label setting.

| Metric (%) | Hierarchical Clustering threshold | | |
| --- | --- | --- | --- |
| | 0.9 | 0.95 (seed 0) | 0.85 (seed 0) |
| Recall | $62.0 \pm 3.7$ | **63.9** | 62.7 |
| Precision | $39.1 \pm 2.4$ | **41.7** | 36.2 |
| F1 | $44.1 \pm 2.4$ | **47.2** | 40.4 |
| AUC | $85.0 \pm 1.7$ | 86.4 | **86.6** |

- High-confidence concepts: expressions that they were certain represented a concept.
- Low-confidence concepts: expressions they were unsure represented a concept.

The study was restricted to captions only, without access to the corresponding images. Participants could choose one of two annotation methods:

- Direct annotation in Excel/CSV: Participants added two new columns (High confidence concepts and Low confidence concepts) to the provided file and returned the completed version.

- Streamlit annotation tool: Alternatively, participants could use a lightweight tool we provided. After setting up the environment and running the script, captions were presented one by one. Concepts were extracted into High and Low fields, and progress was saved automatically. Upon completing the 50 captions, a submission file was generated.

The instructions emphasized that "digit" and "background" were not to be considered concepts. Each concept was defined as a word or phrase that could describe an aspect of the image. An example was also provided:

Caption: The image shows a small digit with moderate thickness, which has undergone a minor translation to the right.

Concepts: small, moderate thickness, minor translation right

Participants could also add optional notes to justify their choices.

**Results & Discussion.** The inter-annotator agreement analysis revealed important differences between high (Figure 5) and low confidence concepts (Figure 6), where SCS-Recall was computed using one annotator as reference and another as query. As expected, agreement was consistently higher for high confidence annotations, reflecting that participants identified transformations (e.g., flips), textures, and colors as objectively measurable and thus reliable. In contrast, agreement was lower for low confidence concepts, which participants frequently justified as subjective or ambiguous. These included relative quantifiers (e.g., "small," "moderate," "minor"), which cannot be precisely measured; absence descriptors (e.g., "featureless," "unremarkable"), which describe the lack of a characteristic rather than a feature itself; and abstract concepts (e.g., "spring-like appearance"), which require external knowledge or encapsulate multiple attributes. Notably, when both high and low confidence concepts (Figure 7) were considered together, agreement scores surpassed those of high-confidence concepts alone. This suggests that participants often recognized the same underlying concepts but diverged in the degree of confidence with which they categorized them.

Two participants (2 and 9) were familiar with the work; however, this did not significantly impact the annotation process, as participants without prior knowledge still achieved high agreement and recall with these annotations.

The results in Table 15 further confirm the inter-annotator findings: participants generally agreed on high-confidence concepts, while low-confidence concepts were more difficult to assess due to their subjective or abstract nature and variability in participant interpretation. Additionally, when comparing against the dataset's ground truth concepts, participants 2 and 9 achieved recalls of 88.2% and 85.3%, and precisions of 87.4% and 89.3%, respectively, showing that CELF performs on par with expert annotators (see Table 2).

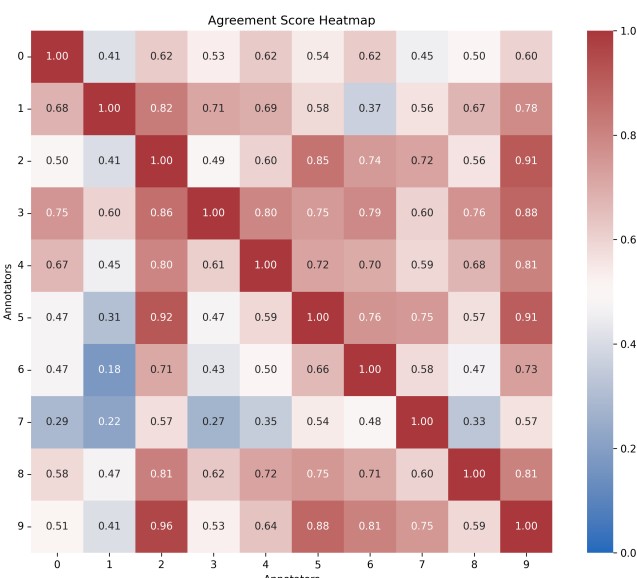

Figure 5: Heatmap of SCS-Recall when considering high confidence concepts. Rows correspond to references (y-axis) and columns correspond to queries (x-axis).

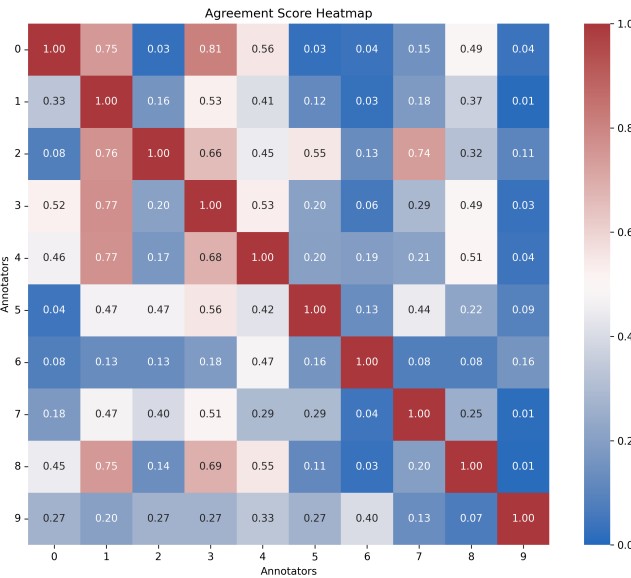

Figure 6: Heatmap of SCS-Recall when considering low confidence concepts. Rows correspond to references (y-axis) and columns correspond to queries (x-axis).

Table 15: Evaluation of concept annotation on the CCM dataset. We report SCS Recall (SCS-R), Precision (SCS-P), and BERTScore Recall (BS-R) for high-confidence, low-confidence, and both concepts.

| CCM Metrics (%) | Both | Low Confidence | High Confidence |
|---|---|---|---|
| SCS-R | **82.8 ± 4.6** | 21.0 ± 16.6 | 62.0 ± 18.3 |
| SCS-P | 68.6 ± 8.7 | 35.7 ± 26.8 | **72.6 ± 12.2** |
| BS R | **95.7 ± 0.4** | 69.6 ± 22.1 | 93.2 ± 2.6 |

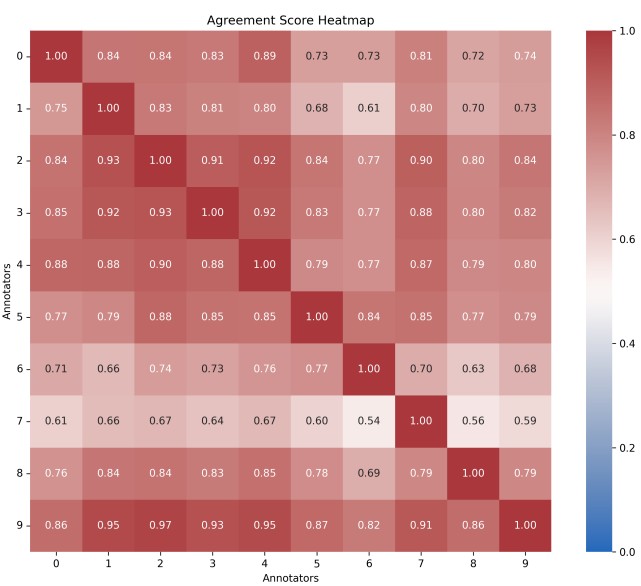

Figure 7: Heatmap of SCS-Recall when considering both high and low confidence concepts. Rows correspond to references (y-axis) and columns correspond to queries (x-axis).

## G.6 QUALITATIVE ANALYSIS OF EXTRACTED CONCEPTS

Figure 8 illustrates examples of extracted concepts from multiple methods applied to the same set of input image-caption pairs. These qualitative comparisons highlight distinct behaviors and failure modes across the approaches.

CELF and CLIP-attn generally exhibit strong alignment with ground truth concepts, occasionally merging multiple concepts into a single one or introducing a small number of irrelevant words. Despite these minor issues, the extracted concepts are predominantly relevant.

In contrast, the pseudo-labels used to fine-tune CELF show notable instability. While it sometimes has useful concepts, it also has unrelated or overly generic terms. As seen in the final row of Figure 8, the pseudo-labels are largely misaligned with the caption content, reflecting its known tendency to hallucinate.

FALCON* tends to extract incomplete or fragmented concepts, largely due to its reliance on noun phrase chunking. Many of its outputs are either too generic or disconnected from the core semantics of the visual and textual input. GRAD-ECLIP shares this limitation: its noun phrase-driven extraction mechanism frequently yields noisy or partial concepts.

Overall, this qualitative evaluation reinforces the need for methods that balance visual grounding with structured concept representation. Among the compared models, CELF with and without fine-tuning (CLIP-attn) most consistently identifies relevant and interpretable concepts aligned with the ground truth concepts.

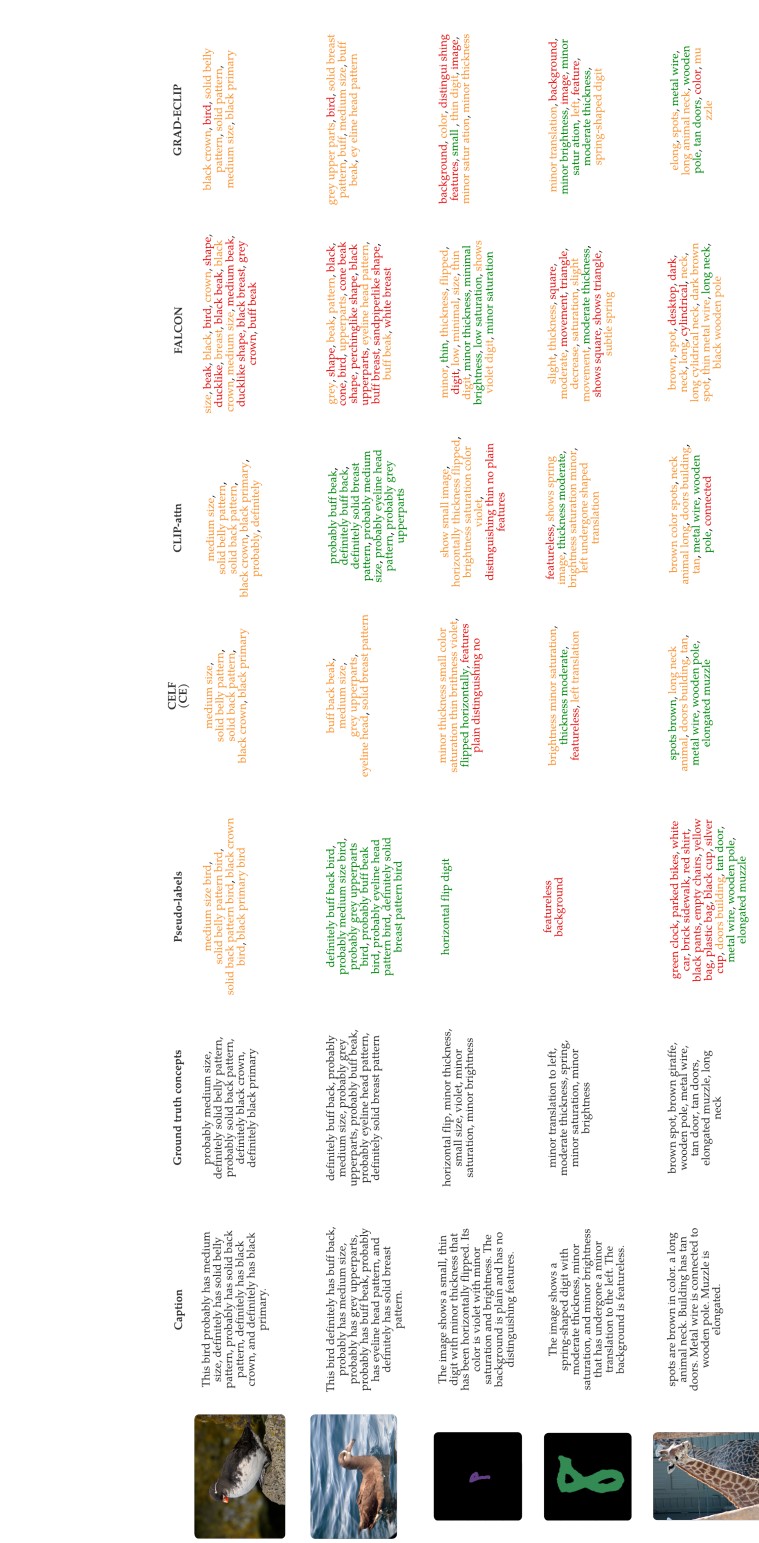

Figure 8: Extracted or generated concepts from each method. The first two images are from the CUB dataset, the next two from C-MNIST, and the last one from VG. Green indicates an exact match with the ground truth concepts; yellow indicates a partial match (some noisy or missing relevant words); red indicates completely irrelevant words. Note that these images are shown for illustration only and are not used during CELF's inference process.

## G.7 DOWNSTREAM TASK

We additionally evaluate a cross-attention CLIP with a linear layer to facilitate a direct comparison with XCB (Alukaev et al., 2023). XCB shows substantially lower performance, likely reflecting its design trade-offs that prioritize interpretability over predictive accuracy. Furthermore, our clustering mechanism effectively groups synonymous concepts, enhancing semantic coherence, which XCB lacks.

VLG-CBM performs better, likely because its concepts are cleaner and annotated at the image level. However, many generated concepts are unrelated to the dataset context or label, highlighting a key limitation of relying on LLMs to produce class-specific concepts.The results also suggest that the weaker performance of the cross-attention CLIP classifier with retrieved concepts adversely affects downstream task outcomes. A promising direction for future work is to assess concept bottleneck models using our extracted concepts instead of LLM-generated ones.

Table 16: Performance of using only concept information on the CCM dataset (macro-averaged Recall, Precision, F1-Score, and AUC), we compare a classifier using RC and EC, and XCB and VLG-CBM, existing CBMs.

| Dataset | Metric (%) | Classifier w/ RC | Classifier w/ EC | XCB | VLG-CBM (seed 0) |
|---------|-----------|------------------|------------------|-----|------------------|
| CCM | Recall | **97.6 ± 3.2** | 81.3 ± 2.4 | 49.2 ± 2.6 | 88.9 |
| | Precision | 51.0 ± 22.9 | 73.0 ± 7.7 | 57.4 ± 7.0 | **78.9** |
| | F1-Score | 62.5 ± 17.2 | 74.4 ± 5.5 | 45.9 ± 2.6 | **82.7** |
| | AUC | 62.9 ± 18.6 | 89.2 ± 3.1 | 85.4 ± 1.2 | **92.2** |

We provide four heatmaps to analyze the concept extraction behavior across labels, their relevance to the downstream task, and alignment with ground truth annotations.

Figure 9 reveals that several control-based labels, such as "off-center," "bright," and "translation left," show high-frequency associations with specific concepts (frequency > 0.5). These patterns suggest that the extracted concepts for these labels are well-aligned and discriminative. In contrast, other labels do not show consistently high-frequency concept associations. This is likely due to their relative scarcity in the dataset, resulting in reduced statistical reliability in concept co-occurrence, and the presence of several variances in the extracted concepts. Additionally, the concept "moderate thickness," being the most frequent across the dataset, shows high co-occurrence across nearly all labels, indicating limited discriminative power and potential redundancy.

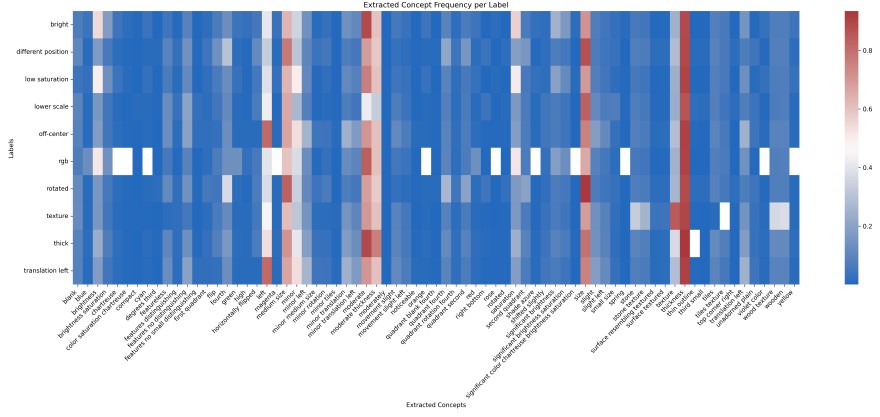

Figure 9: Heatmap of Concept Frequency per Label.

Figures 10 and 11 show that the model often attributes high relevance to semantically aligned concepts (e.g., "left" being dominant for the "translation left" label). However, some unrelated concepts receive

high relevance due to their frequency or semantic overlap in clustering. These observations also highlight the occasional challenges in separating semantically close but distinct concepts, especially when quantifiers are involved. Our findings underscore the benefits of LLM-guided refinement in improving clustering granularity and mitigating such issues.

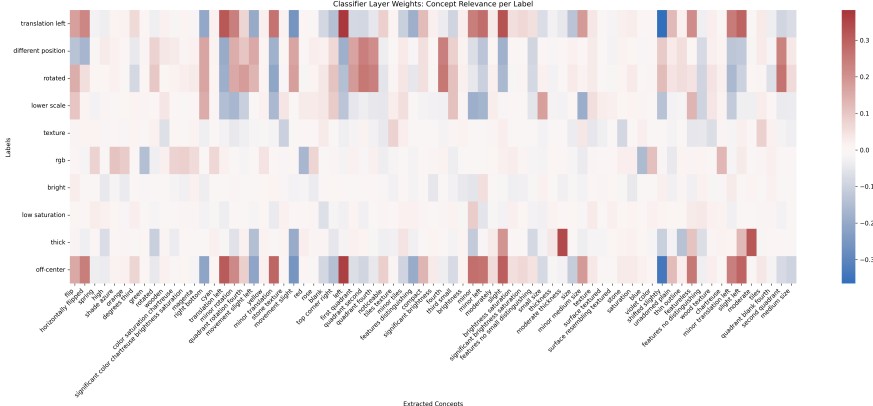

Figure 10: Heatmap displaying model-assigned importance of concepts per label for model trained with image features and extracted concepts.

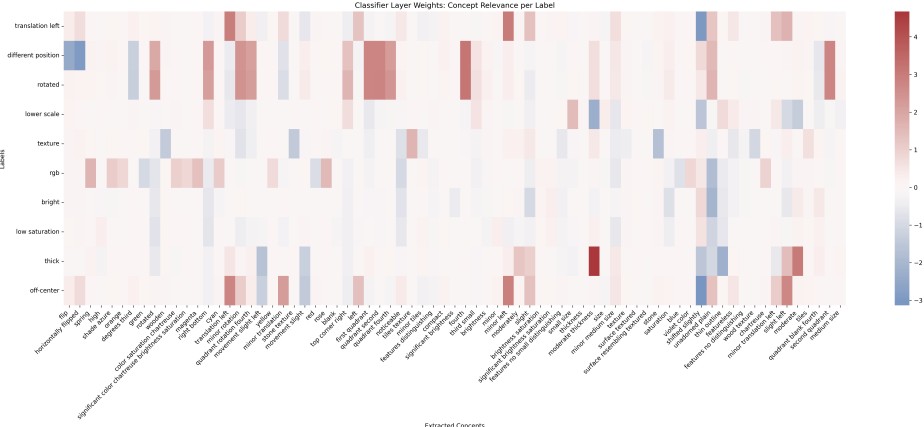

Figure 11: Heatmap displaying model-assigned importance of concepts per label for model trained with only extracted concepts.

Figure 12 further supports these trends. It shows strong hierarchical alignment between extracted and annotated concepts (e.g., "size" frequently co-occurs with "medium size"), alongside occasional overlap between contradicting concepts, revealing the nuanced nature of quantifier handling in automatic concept extraction.

Overall, these visualizations validate the relevance and interpretability of the extracted concepts across labels, while also offering diagnostic insight into clustering granularity and frequency-driven biases. These are aspects where further refinement, particularly in the clustering process, remains a promising direction. They demonstrate the robustness of our method and highlight the value of concept-level analysis for both qualitative and quantitative evaluation.

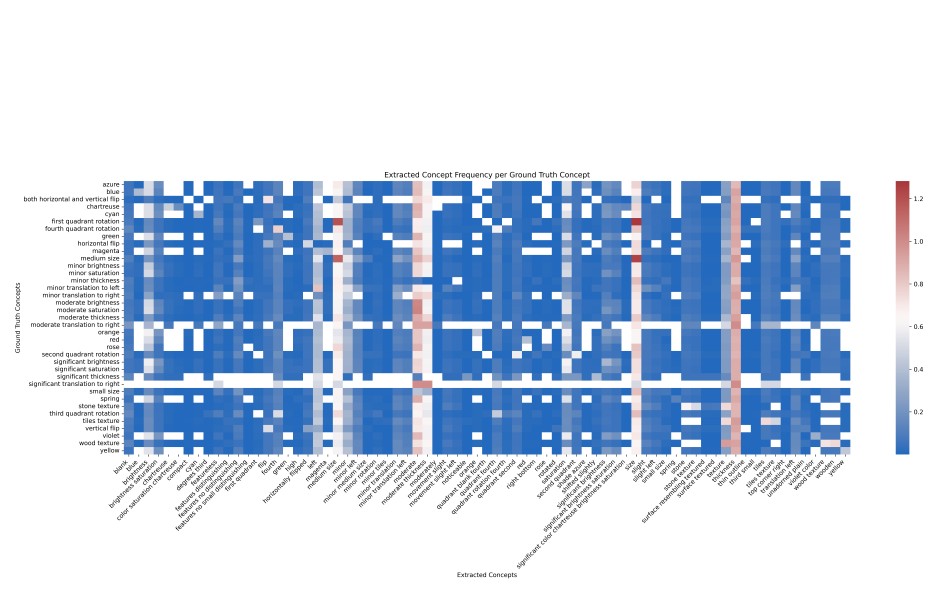

Figure 12: Heatmap of Extracted vs. Ground Truth Concepts.

## H   COMPUTATIONAL RESOURCES

Experiments were run using a GPU partition with NVIDIA L40S and a maximum of 12 CPUs. Due to the modular nature of our pipeline and dataset-dependent variability, runtimes varied across stages:

- LLM-Based Concept Generation (CUB dataset): ~8 hours to generate concepts for 15,000 samples, with runtime dependent on sample complexity and LLM token limits.
- CLIP Fine-Tuning for Concept Extraction (CUB dataset): ~4 hours and 30 minutes.
- Concept Retrieval (C-MNIST dataset):
    - Stage 1 (contrastive alignment): ~5 hours and 30 minutes.
    - Stage 2 (concept prediction): ~6 hours.

We report the peak GPU memory consumption and runtime for all evaluated methods and datasets. For all experiments except those involving only LLMs, we used NVIDIA L40S GPUs. LLM-based methods were run on a V100 GPU. The CE step required between 14-16 GB across all datasets during training and around 4 GB during inference. Among all components, the most memory-intensive stage is the first phase of concept retrieval, where we fine-tune CLIP. Notably, this phase accounts for the highest peak GPU memory usage (36 GB on C-MNIST); however, employing mixed precision lowers it to 26 GB. This presents a scalability bottleneck, particularly for datasets with a large number of concepts or diverse visual-textual pairs. A promising avenue to mitigate this issue is to restrict fine-tuning to the top CLIP layers only. In contrast, the second stage of CELF is significantly lighter. For instance, this stage requires less than 4 GB of memory and runs in under 16 minutes per epoch. CELF's concept extraction module also has relatively low inference memory requirements ( 3.5 GB) and fast runtimes (7 minutes in CUB and 34 seconds in CMNIST).

Compared to baselines, CELF is competitive:

- LLM-based methods consume low memory ( 4.5 GB) but are slower, taking approximately 12 minutes per batch of 128 samples.
- GRAD-ECLIP requires 1.6-10 GB of memory, with runtimes between 4 and 10 minutes.
- FALCON exhibits high memory and time requirements on CUB (24.3 GB and over 14 hours).

Overall, CELF balances training cost and inference efficiency. While the initial step of the CR stage is computationally demanding, the remainder of the pipeline is lightweight.

## I   LLM USAGE

LLMs were used as part of the framework, contributing to concept generation for CELF and caption generation for C-MNIST. They were also used to polish text and correct grammatical errors.