# OpenReview forum: "CELF: A Self-Supervised Multimodal Framework for Concept-Based Interpretability"
_ICLR.cc/2026/Conference — Submitted to ICLR 2026_

### Official Review · Reviewer_cYEd · 2025-10-28

**Soundness:** 3
**Presentation:** 3
**Contribution:** 2
**Rating:** 6
**Confidence:** 4

**Summary:**

This paper introduces CELF, a self-supervised framework to extract human-interpretable concepts from image-caption pairs. The goal is to address the unreliability of concepts generated by LLMs for XAI. The method involves using an LLM to generate pseudo-labels from captions, which then supervise a fine-tuned CLIP model to perform concept extraction. The authors also contribute a new synthetic benchmark C-MNIST and an evaluation metric SCS.

**Strengths:**

- The core technical idea is a novel self-supervised approach that fine-tunes a CLIP-based concept extractor using virtual labels generated by LLM from free-text captions. This approach is valid and has been adequately tested.

- The lack of standardized benchmarks is a known problem in concept-based XAI. The introduction of the C-MNIST dataset and the SCS metric are valuable contributions to the field.

**Weaknesses:**

- The combination of LLM and CLIP was said to be unexplored, but it seems that Kim et al.[1] addressed the problem similarly in their paper.

- As this is a paper about XAI, it seems that there should be results to see if it can actually be utilized in places where XAI is needed, such as the medical domain.

[1] Kim, Injae, et al. "Concept bottleneck with visual concept filtering for explainable medical image classification." International Conference on Medical Image Computing and Computer-Assisted Intervention. Cham: Springer Nature Switzerland, 2023.

**Questions:**

- Please explicitly discuss the relationship between CELF and Kim et al [1].

- It would be nice to see some qualitative results to show how useful it is in reality.

---

> ### Author Response · Authors · 2025-11-19
> **Clarifications and Responses to Reviewer Comments**
>
> We thank reviewer cYEd for the constructive feedback. Below, we provide our responses to the comments.
>
> **W1 & Q1.** We thank the reviewer for pointing out Kim et al. [1]. Our claim that combining LLMs and VLMs for concept-based XAI is "underexplored" was too strong in light of this and related work. Kim et al. generate class-level concepts with an LLM and filter them using visual evidence for medical image classification. In contrast, CELF is caption-driven: we extract concepts from per-image captions and use them as pseudo-labels to train the text encoder to identify caption-level concepts, which are then aligned with image regions. This avoids mistaken associations between class-level concepts and specific images, but requires captions for the extraction phase.
>
> We revised Section 5 and added Appendix A (with additional related works) to (i) explicitly cite Kim et al. [1], (ii) soften our claim to "less explored for caption-level concept extraction from image-text pairs," and (iii) clarify the distinction between class-driven (Kim et al.) and caption-driven (CELF) concept pipelines.
>
> **W2.** We agree that demonstrating utility in XAI-critical domains such as medical imaging is important. In the current manuscript, we provide two pieces of evidence towards practical usefulness: (i) a downstream classification experiment where adding CELF’s concepts improves performance over an image-text pair baseline (Section 6.3), and (ii) an experiment using only concepts as input (Appendix G.7), showing that CELF’s concept representations can support prediction on their own. We made this connection to XAI applications more explicit in the discussion section and highlighted safety-critical domains, such as medical, as a primary target for future evaluation (we are currently exploring a dataset of medical imaging and associated reports, which we plan to report in follow-up work).
>
> **Q2.** We appreciate this "request" for qualitative results to help assess practical usefulness. We included qualitative examples of extracted concepts and their corresponding images in Appendix G.6, along with representative cases where CELF succeeds and fails, to provide a clearer picture of its behavior in realistic scenarios.
>
> We encourage the reviewer to see responses to related comments provided by other reviewers, which address overlapping points and provide additional clarifications.
>
> **Reference:**
>
> [1] Kim, Injae, et al. "Concept bottleneck with visual concept filtering for explainable medical image classification." International Conference on Medical Image Computing and Computer-Assisted Intervention. Cham: Springer Nature Switzerland, 2023.

---

### Official Review · Reviewer_YU1w · 2025-10-30

**Soundness:** 2
**Presentation:** 1
**Contribution:** 1
**Rating:** 2
**Confidence:** 4

**Summary:**

The paper proposes the CELF framework, a self-supervised approach that utilizes a Large Language Model (LLM) for pseudo-label generation and fine-tunes a CLIP model with an additional keyphrase extraction task to achieve concept extraction and concept retrieval. The authors also introduce a new synthetic dataset called C-MNIST, consisting of transformed digits, and a novel evaluation metric, Semantic Cosine Similarity (SCS).

However, the overall technical originality of the proposed dataset, method, and metric does not offer sufficient novelty or value to warrant acceptance at a top-tier venue at this time. Furthermore, significant concerns regarding the experimental setup and methodological consistency need to be addressed.

**Strengths:**

The proposed dataset C-MNIST is well-configurable and might be useful for controlled benchmarking.

**Weaknesses:**

1. The paper fails to adequately discuss existing work and establish a fair comparative context.

   **Synthetic Datasets**: The discussion regarding synthetic, concept-labeled datasets is incomplete. The paper misses important relevant work, such as the synthetic dataset SUB (ICCV 2025) proposed in [1], which provides 38,400 images combining a base class with a single target attribute modification. A comparative discussion or experimental inclusion of this work is necessary.

   **VLM/LLM Concept Extraction**: The authors limit their comparison primarily to FALCON, XCB, LF-CBM, and Grad-ECLIP, overlooking several crucial and recent contributions in the area of VLM and LLM-based concept extraction and interpretability, such as [2], [3], [4], and [5]. This omission makes the claimed novelty and state-of-the-art comparison questionable.

2. The paper contains a serious internal inconsistency. The authors claim that LLM-based methods suffer from hallucination and VLM-based methods require large-scale pre-training knowledge. Yet, the proposed CELF framework requires fine-tuning CLIP, which itself introduces a substantial computational burden (as noted by the 36 GB GPU usage) and carries the inherent risk of disrupting CLIP's robust, well-generalized semantic space which it was designed to leverage.

3. Given that CELF includes an expensive, domain-specific fine-tuning step on CLIP, comparing its performance against off-the-shelf, non-fine-tuned CLIP-based baselines (such as the implementations of GRAD-ECLIP or LF-CBM methods) presents an unfair comparison.



[1]  SUB: Benchmarking CBM Generalization via Synthetic Attribute Substitutions (ICCV 2025)

[2] Interpreting CLIP with Sparse Linear Concept Embeddings (SpLiCE) (NeurIPS 2024)

[3] V2C-CBM: Building Concept Bottlenecks with Vision-to-Concept Tokenizer (AAAI 2025)

[4] Text-To-Concept (and Back) via Cross-Model Alignment (ICML 2023)

[5] From attribution maps to human-understandable explanations through Concept Relevance Propagation (Nature Machine Intelligence 2023)

**Questions:**

1. Please supplement the paper with a thorough discussion and fair comparison against the related recent work mentioned in Weakness 1. Please provide a detailed explanation of why the comparison was specifically focused on FALCON, XCB, LF-CBM, and Grad-ECLIP, and why these alone constitute a sufficient and representative set of baselines.
2. Please explicitly address the critical methodological inconsistency regarding CLIP fine-tuning. Since CELF requires fine-tuning CLIP, how do the authors ensure this process does not negatively influence the generalization ability of the CLIP feature space? Furthermore, please justify the statement that the method "avoids overfitting risks and retains transparent, easily interpretable attention-based concept scores" when the core VLM weights are intentionally being altered. This fine-tuning paradigm significantly lowers the practical generality and value of the approach.
3. I recommend the authors carefully reconsider the fundamental premise and methodology of this paper, as well as the structural organization of the manuscript for better paper quality.

---

> ### Author Response · Authors · 2025-11-19
> **Clarifications and Responses to Reviewer Comments - Part 1**
>
> We thank reviewer YU1w for the detailed feedback and for recognizing the configurability and potential utility of the C-MNIST dataset for controlled benchmarking. We also appreciate the reviewer’s perspective regarding the novelty and overall contribution. We have added a discussion section to clarify the conceptual motivation, the distinctions from prior work, and the methodological consistency of our evaluation, and will continue to refine the manuscript to ensure these points are clearly conveyed. Below, we provide point-by-point responses to the reviewer’s comments.
>
> **W1 (Synthetic Datasets).** We thank the reviewer for pointing out this resource. SUB [1] is a good benchmark to test whether models truly ground attribute concepts under distribution shift. Its images are generated via a diffusion model and filtered for attribute consistency, whereas C-MNIST is rule-based and designed to provide exact ground truth for both visual and textual concepts under controlled transformations. Because SUB targets CUB-like fine-grained birds, it is complementary to our current evaluation since it would test generalization under attribute substitution rather than extraction fidelity under noisy captions.
>
> While we did not include SUB experiments due to space and compute constraints, we (i) added a discussion of SUB in the synthetic-dataset paragraph of Section 2, explicitly contrasting its goal with C-MNIST, and (ii) highlighted evaluating CELF on attribute substitution benchmarks like SUB as an important extension for future work on concept generalization.
>
> **W1 (VLM/LLM Concept Extraction) & Q1.** We thank the reviewer and agree that these recent methods are relevant to concept-based interpretability. Conceptually, they address related questions with different assumptions:
>
> -	SpLiCE [2] and Text-To-Concept [4] interpret or adapt existing VLMs using a fixed concept dictionary or alignment, but do not learn a structured concept space from captions as CELF does.  SpLiCE extracts frequent words or phrases from LAION captions (similar to FALCON) and ranks concepts using cosine similarity to reduce redundancy, followed by a top-K selection to create the fixed concept dictionary and then operating at the image level.
> -	V2C-CBM [3] builds a vision-to-concept tokenizer over a pre-defined vocabulary, whereas CELF discovers concepts directly from caption data without a fixed list.
> -	Concept Relevance Propagation [5] explains arbitrary networks given manual textual representation of visual concepts, while CELF learns both the concept set and their image alignment automatically.
>
> This makes a per-caption evaluation against caption-derived ground truth concepts challenging and potentially misleading. For instance, on CUB, an image-level method can extract concepts related to background objects that are not mentioned in the caption. Without manual image annotations, it is impossible to tell whether a "missing" concept is a model error or simply absent from the caption. CELF’s evaluation explicitly measures the ability to recover caption-relevant concepts, which are closer to the human-provided supervisory signal and less affected by background noise.
>
> Our baseline selection was guided by the setting we’re studying: per-caption concept extraction and retrieval from image-text pairs, where ground truth  concepts are derived from captions rather than from class labels or manually specified concept sets. FALCON, XCB, and Grad-ECLIP were chosen because they can be adapted to this caption-conditioned setting:
>
> -	FALCON\* and Grad-ECLIP extract words from captions and operate or can be adapted to operate directly on image-text pairs.
> -	XCB learns concept queries from image-text pairs and can be evaluated on C-MNIST’s concept labels.
>
> LF-CBM is discussed in the related work because it relies on LLM-generated concepts. However, since XCB reports stronger performance than LF-CBM, we use XCB as the quantitative baseline for comparison with CELF.
>
> We view classification-based CBM evaluations as complementary. They assess the downstream utility of concept representations, whereas our main focus in this paper is the semantic correctness of extracted concepts themselves (quantified via SCS and human studies).
>
> In the revised manuscript we i) extended the related work in Appendix A to explicitly discuss SpLiCE, V2C-CBM, Text-To-Concept, and Concept Relevance Propagation, and ii) clarified why our main quantitative comparisons focus on caption-level methods in Section 2 and 7 (a new Discussion section).

---

> > ### Author Response · Authors · 2025-11-19
> > **Clarifications and Responses to Reviewer Comments - Part 2**
> >
> > **W2, W3 & Q2.** We apologize for the confusion caused by our phrasing and thank the opportunity to clarify our fine-tuning  strategy and its implications.
> >
> > Our statement that LLM-based methods suffer from hallucination refers specifically to our observations that the LLM occasionally produced concepts not present in the caption or copied artifacts from the prompt examples. This motivates a learning-based mechanism to correct and filter pseudo-labels. Similarly, our comment that VLM-based methods require large-scale pretraining knowledge targeted at off-the-shelf CLIP used without adaptation, which can result in their performance dropping in out-of-domain data. As shown in Appendix Table 11, a similarity-only CLIP baseline underperforms on C-MNIST, likely due to the domain gap. CELF’s fine-tuning is not intended to replace CLIP’s semantic space but to adapt it to the caption structure and concept supervision of the target dataset.
> >
> > Regarding the concern about disrupting CLIP’s feature space and fairness of the comparison:
> >
> > -	*How we fine-tune CLIP and preserve its structure:* In the CE task, the objective explicitly includes CLIP’s original image-text contrastive loss, combined with a multi-label concept-supervision loss. This joint objective is designed to reinforce CLIP’s multimodal structure rather than replace it. In practice, the image encoder largely preserves its pretrained representations, while the text encoder is adapted to better identify concept-relevant tokens. Importantly, we do not add a new learnable classifier or concept-specific attention head. The attention mechanism used to produce concept scores remains CLIP’s native attention. Fine-tuning changes the underlying representations under a supervised signal, but the computation and transparency of the attention-based scores are unchanged. Concept scores stay directly traceable to CLIP’s internal attention patterns.
> > -	*Empirical evidence that CLIP’s fine-tuning is not essential for CELF to work:* CELF also supports a fully frozen-CLIP configuration for CE (CLIP-attn).  As shown in Table 2, this variant already outperforms off-the-shelf CLIP-based baselines such as Grad-ECLIP and FALCON on concept extraction. Thus, CELF does not rely on fine-tuning to function: fine-tuning is a task-driven enhancement that further improves concept separation and attention stability. For CR, we additionally report results with a frozen CLIP backbone (updated Table 3). In this setting, recall improves but precision and F1 drop by more than 5 p.p., consistent with CLIP being originally optimized for global image-caption alignment rather than fine-grained concept-image alignment, and with the domain gap between C-MNIST and CLIP’s pretraining data. This confirms that our architectural modifications (graph-guided concept retrieval) already add value on top of CLIP, and that fine-tuning further adapts CLIP to the specific CR task.
> > -	*The most memory-intensive stage is the CR fine-tuning, not CE:*  This stage only affects comparison to XCB, which is also trained. We have additionally reduced the peak memory from 36 GB to 26 GB by using mixed precision.
> > -	*Why we claim reduced overfitting risk and retained interpretability:* to limit overfitting and preserve interpretability, we constrain capacity in two ways: (i) we intentionally avoid introducing a separate learnable head (e.g., MLP or linear classifier) for concept prediction, which would increase the risk of memorizing noisy pseudo-labels and decoupling concept scores from CLIP’s internal attention; (ii) we regularize fine-tuning by jointly optimizing CLIP’s original contrastive loss, encouraging the model to preserve its general image-text embedding structure while improving alignment with task-specific concepts.
> >
> > We clarified in Sections 5, 6, and 7 that (i) a frozen-CLIP variant of CELF already performs well, (ii) fine-tuning is optional but beneficial when domain adaptation is expected to yield measurable improvements in concept quality, (iii) this fine-tuning remains aligned with CLIP’s original training objective and preserves transparent attention-based concept scores, and (iv) we explicitly separate comparisons involving trained models (CELF, XCB) from those relying on off-the-shelf CLIP (CLIP-attn, FALCON, Grad-ECLIP) to ensure a clear evaluation protocol.

---

> ### Author Response · Authors · 2025-11-19
> **Clarifications and Responses to Reviewer Comments - Part 3**
>
> **Q3.** We thank the reviewer for the recommendation. Our premise is that many existing concept-based methods either require manual concept labels or rely solely on LLM-generated concept sets, and that a self-supervised, caption-driven framework can improve both scalability and semantic fidelity. We will make this premise more explicit in the introduction.
>
> To improve clarity in the revised manuscript, we now clearly distinguish:
>
> -	the problem statement (caption-level concept extraction and retrieval from image-text pairs);
> -	the motivation for our CE fine-tuning approach (rephrased in the main text using the explanation provided in the previous response);
> -	the role of C-MNIST, positioned as a controlled benchmark for evaluating concept fidelity rather than as a competing CBM dataset.
>
> To further strengthen the structure, we (i) moved some technical details (e.g., clustering hyperparameters) to the appendix to streamline the main text, and (ii) expanded the related work in Appendix A to more clearly situate CELF relative to recent CBM and VLM/LLM-based methods mentioned in Weakness 1 and Question 1. We will continue refining the methods section for readability, and we hope these revisions make the scope and contribution of our work clearer.
>
> We encourage the reviewer to see responses to related comments provided by other reviewers, which address overlapping points and provide additional clarifications.
>
> **References:**
>
> [1] SUB: Benchmarking CBM Generalization via Synthetic Attribute Substitutions (ICCV 2025)
>
> [2] Interpreting CLIP with Sparse Linear Concept Embeddings (SpLiCE) (NeurIPS 2024)
>
> [3] V2C-CBM: Building Concept Bottlenecks with Vision-to-Concept Tokenizer (AAAI 2025)
>
> [4] Text-To-Concept (and Back) via Cross-Model Alignment (ICML 2023)
>
> [5] From attribution maps to human-understandable explanations through Concept Relevance Propagation (Nature Machine Intelligence 2023)

---

### Official Review · Reviewer_CTq3 · 2025-11-02

**Soundness:** 3
**Presentation:** 3
**Contribution:** 2
**Rating:** 2
**Confidence:** 4

**Summary:**

The paper proposes an automated method for extracting human-interpretable concepts from vision-language data for concept-based explainability methods, which has been limited primarily by manual efforts and LLM hallucinations.

**Strengths:**

1. The paper demonstrates superior concept extraction on the proposed C-MNIST dataset

**Weaknesses:**

1. The motivation for introducing C-MNIST to measure concept-extraction performance, rather than directly evaluating Concept Bottleneck Models (CBMs) [1, 2, 3, 4] with generated concepts, is unclear. Further, the authors also fail to provide any meaningful experiments or examples to demonstrate that the C-MNIST benchmark is superior to directly evaluating CBM approaches in terms of accuracy.
2. The authors compare Concept Extraction performance with a very limited baseline and lack comparison with concept generation methods that utilize both language and vision data. Recent methods, including [4], have demonstrated improvement in LLM-generated concept sets by grounding with an open-vocabulary detection model.
3. The paper lacks qualitative results demonstrating the generated concepts on standard datasets, such as CUB and CIFAR, which is crucial for demonstrating the performance of the proposed method in real-world settings.
4. Formatting issues: Fig. 2, Fig. 9, and Fig.10 have a really small font size

[1] Yang, Yue, et al. "Language in a bottle: Language model guided concept bottlenecks for interpretable image classification." Proceedings of the IEEE/CVF conference on computer vision and pattern recognition. 2023.

[2] Oikarinen, Tuomas, et al. "Label-free concept bottleneck models." arXiv preprint arXiv:2304.06129 (2023).

[3] Yan, An, et al. "Learning concise and descriptive attributes for visual recognition." Proceedings of the IEEE/CVF International Conference on Computer Vision. 2023.

[4] Srivastava, Divyansh, Ge Yan, and Lily Weng. "Vlg-cbm: Training concept bottleneck models with vision-language guidance." Advances in Neural Information Processing Systems 37 (2024): 79057-79094.

**Questions:**

My primary concerns are the lack of robust evaluation and the missing results on standard datasets. Please see weaknesses for more details.

---

> ### Author Response · Authors · 2025-11-19
> **Clarifications and Responses to Reviewer Comments**
>
> We thank reviewer CTq3 for the detailed feedback. We appreciate the points raised regarding evaluation, baseline comparisons, and qualitative results. Below, we provide point-by-point responses to the reviewer’s comments.
>
> **W1.** We agree that CBMs are a natural reference point. However, their task accuracy alone does not measure whether concepts are semantically correct. Prior work [4] shows that CBMs can achieve good performance even when the bottleneck concepts are random, hinting that accuracy is compatible with non-interpretable concepts. C-MNIST fills a complementary role since it provides ground truth concepts and controlled transformations, allowing us to quantitatively measure extraction accuracy and semantic fidelity (as we do in Table 2 and Appendix G.5), independently of classification performance.
>
> In the revised version, we made this motivation explicit in Section 7 (a new Discussion section) and clarified in Section 3 that C-MNIST can serve as a benchmark for concept extraction quality (and additionally for tasks including retrieval, concept disentanglement and compositionality, etc.), not a replacement for CBM-based evaluation on downstream accuracy.
>
> **W2.** We appreciate the suggestion to compare with CBM variants that combine language and vision [1–4]. Our current baselines (FALCON\*, XCB, Grad-ECLIP) were chosen because they operate directly on image–caption pairs   (with FALCON\* being an easily adapted variant of the original FALCON), which aligns with our setting where ground truth concepts come from captions rather than class labels. Methods such as LaBo [1], LF-CBM [2], and VLG-CBM [4] generate class-conditioned concept sets and then train CBMs. Adapting them to caption-conditioned, per-image extraction would require substantial redesign (e.g., replacing class definitions with captions and changing the training target from class probabilities to concept sets).
>
> We clarified this distinction in Section 7 and Appendix A (an extension of related works), including a short discussion explaining how VLG-CBM’s grounded filtering would behave in our setup (increasing precision at the cost of recall). As mentioned above, we also agree that evaluating CBM accuracy with generated concepts is complementary to our focus on concept fidelity.
>
> **W3 & W4.** Qualitative examples are important, indeed, and we thank the reviewer for the suggestion. We added qualitative visualizations of extracted concepts on C-MNIST, Visual Genome, and CUB to in Appendix G.6 (and increased font sizes in Figures 1/9/10), and clarified in Section 7 that benchmarks such as CIFAR do not include captions and would require an  additional caption generation step to be compatible with CELF’s CE stage. We could, however, use CIFAR image-only inputs in the CR stage and the extracted concepts would come from other datasets.
>
> We encourage the reviewer to see responses to related comments provided by other reviewers, which address overlapping points and provide additional clarifications.
>
> **References:**
>
> [1] Yang, Yue, et al. "Language in a bottle: Language model guided concept bottlenecks for interpretable image classification." Proceedings of the IEEE/CVF conference on computer vision and pattern recognition. 2023.
>
> [2] Oikarinen, Tuomas, et al. "Label-free concept bottleneck models." arXiv preprint arXiv:2304.06129 (2023).
>
> [3] Yan, An, et al. "Learning concise and descriptive attributes for visual recognition." Proceedings of the IEEE/CVF International Conference on Computer Vision. 2023.
>
> [4] Srivastava, Divyansh, Ge Yan, and Lily Weng. "Vlg-cbm: Training concept bottleneck models with vision-language guidance." Advances in Neural Information Processing Systems 37 (2024): 79057-79094.

---

### Official Review · Reviewer_MR8J · 2025-11-03

**Soundness:** 4
**Presentation:** 3
**Contribution:** 3
**Rating:** 6
**Confidence:** 3

**Summary:**

This paper presents a self-supervised approach to extract and align interpretable visual concepts from image–text pairs without manual labels. It introduces CELF, which uses attention-guided keyword extraction and contrastive learning to connect textual concepts with image regions. It also introduces C-MNIST, a configurable dataset providing ground-truth concepts for evaluation. CELF achieves superior concept-extraction accuracy and interpretability compared to existing methods such as FALCON and GRAD-ECLIP.

**Strengths:**

1. It proposes a self-supervised multimodal framework for concept extraction without manual labels, combining attention and contrastive learning to advance interpretable vision–language modeling.
2. It integrates a full pipeline with the CELF framework, the C-MNIST dataset, and the SCS metric, ensuring reproducibility and objective evaluation.
3. It achieves superior concept-extraction accuracy and interpretability across both synthetic and real-world datasets compared to existing baselines.

**Weaknesses:**

1. The framework is mainly evaluated on datasets with clear visual–text correspondences, and its performance on more complex or abstract domains remains unverified.
2. CELF relies heavily on CLIP and LLM-generated pseudo-labels, which may introduce bias or noise from those pretrained components.

**Questions:**

How well would CELF generalize to domains where image–text alignment is noisy or abstract, such as medical or scientific data, and can the framework be adapted to handle such cases effectively?

---

> ### Author Response · Authors · 2025-11-19
> **Clarifications and Responses to Reviewer Comments**
>
> We thank reviewer MR8j for the positive feedback and for highlighting the strengths of our self-supervised concept extraction and retrieval framework. Our detailed responses to your comments are provided below.
>
> **W1 & Q1.** We acknowledge the importance of considering more complex domains since they are particularly important for XAI. Concept Extraction in CELF relies primarily on textual structure, using image-text alignment only as an auxiliary signal, which is why it already works on less structured data such as Visual Genome. For Concept Retrieval, the backbone can be replaced with a domain-specific VLM, which we started exploring in a medical setting. We clarified in Section 7 (a new Discussion section) that i) CELF does not require perfectly aligned captions for CE, but ii) retrieval quality is ultimately bounded by concept quality and domain-specific pretraining, making specialized backbones a natural extension in more complex domains. Additionally, despite the domain-specific aspect, CELF’s training objective opens the possibility of learning broadly transferable foundational concepts in future work (with scaled data).
>
> **W2.** We agree that relying on pretrained CLIP and LLM pseudo-labels introduces a potential source of bias and noise. In CELF, we address this in two complementary ways:
>
> 1– Filtering hallucinations via cross-modal agreement: the LLM is only used to propose candidate concepts from captions. Concepts that are not grounded in the caption or not supported by CLIP’s attention/similarity never enter the training signal.
>
> 2 – Lexical debiasing: high-frequency but semantically unhelpful words (e.g., region-related terms, stopwords) are explicitly removed before graph construction, stabilizing Louvain clustering and preventing them from dominating multiple concepts.
>
> Empirically, our ablations (Appendix G.2.2, Table 11) show that attention-based extraction with fine-tuned CLIP substantially reduces noise compared to similarity-based extraction, and the human study in Appendix G.5 confirms that CELF’s concepts align with human annotations. We clarified this rationale and the empirical evidence in Section 5.1 and Section 7.
>
>
> *Note:* We note that the framework figure in the previous manuscript did not fully correspond to the implementation described in the text: the linear layers corresponding to CLIP’s projection layers were depicted in the second stage, whereas in the implementation they appear in the first training stage. This has been corrected in the revised figure, and we apologize for the oversight.
>
> We encourage the reviewer to see responses to related comments provided by other reviewers, which address overlapping points and provide additional clarifications.

---

### Author Response · Authors · 2025-12-03
**Final Remarks (Summary of Key Points and Changes)**

To facilitate the work of the AC in reviewing this submission, we summarize the main points raised by the reviewers and how we address them. Nonetheless, we invite the AC to review each individual response for more details. Across the reviews, several common themes emerged:

1. The importance of evaluating the framework in more complex domains, such as clinical settings, where concept-based interpretability could have significant impact.

2. The need to acknowledge that relying on pretrained CLIP and LLM-based pseudo-labels may introduce bias or noise.

3. The suggestion to better connect our work to prior concept-based interpretability literature and to clarify baseline choices.

4. A few phrasing ambiguities identified by reviewers, which we have now revised in the manuscript.

Below, we succinctly address these points:

1. We have started applying the framework to clinical data. The primary reliance of CELF's CE on textual structure suggests its effectiveness when paired with an appropriate LLM. Conversely, CR can leverage domain-specific VLMs.
2. CELF incorporates two mechanisms to mitigate noise from pretrained models: (1) cross-modal filtering, which prevents hallucinated concepts from entering training, and (2) removal of irrelevant terms before graph construction. Additionally, our ablations (Appendix G.2.2) and human study (Appendix G.5) support the effectiveness of attention-based extraction.
3. CBMs primarily measure task accuracy, which does not guarantee semantic correctness of concepts; prior work [1] shows that CBMs can achieve good accuracy even with random bottleneck concepts. Moreover, most CBMs operate with class-conditioned concepts, while CELF performs caption-conditioned extraction for each image-text pair. Other methods, such as SpLiCE [2], operate primarily on the image rather than on the caption. For datasets like CUB, image- or class-level approaches may output concepts unrelated to the caption but present in the image or associated with the class. These factors make direct comparison difficult when ground truth is derived from captions. Our baseline selection therefore reflects our evaluation setting, which centers on caption-relevant concepts.

Additionally, we experimented with VLG-CBM [1], which followed LF-CBM [3] approach for concept generation, in our downstream setting and explored the limitations of LLM-generated concept sets: these depend on the LLM’s prior class knowledge and are insensitive to dataset context. For instance, in C-MNIST, GPT-3.5 generated concepts such as "book" or "dictionary" for the class "translation left," incorrectly assuming a linguistic context, or unrelated concepts such as "coffee mug" or "chair" for "different positions." These errors highlight the advantage of CELF as a dataset-specific, caption-grounded concept extraction framework. In CELF, the LLM serves only to extract structures already present in captions, rather than generating concepts based on its knowledge, avoiding these class-dependence issues. The results (updated Table 4) show that our method outperforms VLG-CBM and further support that CBMs trained with LLM-generated concepts can achieve good performance even when concepts are not meaningful.

In the new version of the manuscript, we added a Discussion section (Section 7), expanded the Related Work (Appendix A), added qualitative examples (Appendix G.6), and clarified the positioning of off-the-shelf baselines versus our variants in the Results section.

Our method is among the few that extract concepts in a caption-grounded manner for vision-language datasets, using image features as auxiliary signals and enabling concept retrieval for image-only samples. The results show that CELF is the strongest approach, as per the SCS metrics and the downstream task.

**References:**

[1] Srivastava, Divyansh, Ge Yan, and Lily Weng. "Vlg-cbm: Training concept bottleneck models with vision-language guidance." Advances in Neural Information Processing Systems 37 (2024): 79057-79094.

[2] Interpreting CLIP with Sparse Linear Concept Embeddings (SpLiCE) (NeurIPS 2024)

[3] Oikarinen, Tuomas, et al. "Label-free concept bottleneck models." arXiv preprint arXiv:2304.06129 (2023).

---

### Meta-Review · Area_Chair_r3z6 · 2026-01-07

**Summary:**

This paper received mixed reviews. The main concerns raised in the reviews are:
1. limited evaluation:
   - only evaluated on datasets with clear visual-text correspondence; performance on complex domains is unclear (`MR8J`); no results on practical XAI domains such as medical domain (`cYEd`).
   - motivation for the newly introduced C-MNIST benchmark is unclear (`CTq3`).
   - insufficient comparison with previous work (`CTq3`, `YU1w`); some of comparisons are unfair (`YU1w`).
   - qualitative results lacking (`CTq3`).
2. potential bias in CLIP/LLM-generated pseudo-labels (`MR8J`).
3. inconsistent claims regarding CLIP fine-tuning (`YU1w`).

Overall, reviewers do find merits in this paper but also raised many questions about the current evaluation. This is a borderline submission that would have benefited from more informative reviewer discussion, which unfortunately was not possible. Based on the current evidence, I'm inclined to reject the submission based on outstanding concerns on the evaluation protocols and comparisons.

**Reviewer Concerns:**

1. Concern #1 is partially addressed by the additional discussions on the new benchmark and the baseline selection as well as the brief qualitative results. Although these additional arguments seem reasonable, the discussion of related work and broader contexts in the paper still requires substantial revision. The authors also suggested that they were exploring experiments in medical domains but did not provided concrete results.
2. Concern #2 is largely addressed by the additional clarification.
3. Concern #3 is partially addressed by the further explanation. However, the arguments are largely based on the empirical results of the existing limited evaluation, whose credibility still remains questionable.

**Reviewer Scores:**

It is difficult to predict how each reviewer would have changed their score for this submission had they been able to participate fully in the discussion. The rebuttal provides compelling arguments for some of the questions but still lacks in offering new evidence to address the concerns on the evaluation.

---

### Decision · Program_Chairs · 2026-01-26

Reject